# ERK3/MAPK6 dictates CDC42/RAC1 activity and ARP2/3-dependent actin polymerization

Katarzyna Bogucka-Janczi[1], Gregory Harms[1,2], Marie-May Coissieux[3], Mohamed Bentires-Alj[3], Bernd Thiede[4], Krishnaraj Rajalingam[1,5]*

[1]Cell Biology Unit, University Medical Center of the Johannes Gutenberg University Mainz, Mainz, Germany; [2]Departments of Biology and Physics, Wilkes University, Wilkes Barre, United States; [3]Department of Biomedicine, University of Basel, Basel, Switzerland; [4]Department of Bioscience, University of Oslo, Oslo, Norway; [5]University Cancer Center Mainz, University Medical Center Mainz, Mainz, Germany

**Abstract** The actin cytoskeleton is tightly controlled by RhoGTPases, actin binding-proteins and nucleation-promoting factors to perform fundamental cellular functions. We have previously shown that ERK3, an atypical MAPK, controls IL-8 production and chemotaxis (Bogueka et al., 2020). Here, we show in human cells that ERK3 directly acts as a guanine nucleotide exchange factor for CDC42 and phosphorylates the ARP3 subunit of the ARP2/3 complex at S418 to promote filopodia formation and actin polymerization, respectively. Consistently, depletion of ERK3 prevented both basal and EGF-dependent RAC1 and CDC42 activation, maintenance of F-actin content, filopodia formation, and epithelial cell migration. Further, ERK3 protein bound directly to the purified ARP2/3 complex and augmented polymerization of actin in vitro. ERK3 kinase activity was required for the formation of actin-rich protrusions in mammalian cells. These findings unveil a fundamentally unique pathway employed by cells to control actin-dependent cellular functions.

*For correspondence: krishna@uni-mainz.de

## Editor's evaluation

The manuscript describes a fundamental study of the atypical MAPK, ERK3, in the activation of RhoGTPase Cdc42 and the formation of actin-rich protrusions and cell migration. The results based on compelling evidence show that ERK3 is required for the motility of tumor cells in vivo, providing a new target for fighting metastasis.

## Introduction

Actin is one of the most abundant and highly conserved proteins with over 95% homology among all isoforms (*Vandekerckhove and Weber, 1979*). In cells, it is present in globular/monomeric form (G-actin), which can polymerize into branched and elongated filamentous forms (F-actin) in a dynamic, spatially, and temporally controlled process (*Skruber et al., 2018*; *Lee and Dominguez, 2010*; *Carlsson, 2006*; *Carlier et al., 1997*). The cytoplasmic actin network constitutes an important part of the cytoskeleton, which not only mechanically supports the plasma membrane and gives the cell its shape, but fulfills a variety of other functions: It regulates velocity and directionality of cell migration, enables intracellular signaling and transport, supports cell division, and more (*Yamaguchi and Condeelis, 2007*; *dos Remedios et al., 2003*). Moreover, by branching and bundling of F-actin filaments the actin cytoskeleton can form unique structures such as lamellipodia and filopodia, which in epithelial cells regulate cell polarization and are crucial for the contact with their environment, while in other

cell types these actin rich structures control motility, chemotaxis, and haptotaxis (*Vasioukhin et al., 2000*; *Raich et al., 1999*; *Millard and Martin, 2008*; *Khurana and George, 2011*; *Bahri et al., 2010*).

To perform its cellular functions, the actin cytoskeleton must be able to rapidly remodel in response to intrinsic and extrinsic cues (*Innocenti, 2018*; *Bugyi and Carlier, 2010*; *Theriot and Mitchison, 1991*). The spatiotemporally defined branching and elongation of the polymerized actin filaments is achieved by activation of nucleation factors including the Actin-related protein 2 and 3 (ARP2/3) complex, formins, and spire (*Rouiller et al., 2008*; *Rohatgi et al., 1999*; *Carlier et al., 1999*; *Chesarone and Goode, 2009*; *Campellone and Welch, 2010*; *Pollard, 2007*; *Schuldt, 2005*; *Buracco et al., 2019*). Actin nucleators facilitate the formation of F-actin architectures, branches, and bundles, giving rise to specific actin-rich protrusions, thus generating the pushing force required for membrane protrusion during migration (*Theriot and Mitchison, 1991*; *Svitkina, 2018*; *Korobova and Svitkina, 2008*). It is well established that ARP2/3-dependent actin polymerization generates branched F-actin network, while formins generate unbranched filaments and promote growth of actin filaments at the barbed end (*Pollard, 2007*; *Wood and Martin, 2002*; *Yang and Svitkina, 2011*). Two models of the filopodia initiation emerged, where ARP2/3 and formins are thought to be involved in the 'convergent elongation' and 'tip nucleation' model, respectively. However, the proposed models are not mutually exclusive and filopodia formation could engage both ARP2/3 and formins (*Yang and Svitkina, 2011*). Moreover, both in vivo and in vitro experimental evidence upholds the essential role for ARP2/3 complex-dependent actin nucelation in the formation of filopodia-like protrusions (*Korobova and Svitkina, 2008*; *Svitkina et al., 2003*; *Machesky and Insall, 1998*; *Falet et al., 2002*; *Vignjevic et al., 2003*; *Haviv et al., 2006*; *Brill-Karniely et al., 2009*; *Lee et al., 2010*).

ARP2/3 complex is a major and highly conserved actin nucleator composed of seven subunits (*Robinson et al., 2001*; *Goley and Welch, 2006*). The structural and biochemical properties of the ARP2/3 complex have been intensively studied in vitro, and apart from ATP, its activation relies on binding to pre-formed actin filament and nucleation-promoting factors (NPFs) (*Robinson et al., 2001*; *Goley and Welch, 2006*; *Pizarro-Cerdá et al., 2017*; *Nolen et al., 2004*; *Nolen and Pollard, 2007*; *Espinoza-Sanchez et al., 2018*). The main regulators of the ARP2/3 complex belong to the Wiskott–Aldrich syndrome protein (WASP) family proteins, including WASP-family verprolin-homologous protein (WAVE) (*Yarar et al., 1999*; *Machesky et al., 1999*; *Higgs et al., 1999*). WASP and WAVE proteins serve as a link between the actin cytoskeleton and RhoGTPases via direct and intermediate binding to cell division control 42 (CDC42) and Ras-related C3 botulin toxin substrate 1 (RAC1), respectively (*Rohatgi et al., 1999*; *Carlier et al., 1999*).

The small RhoGTPases belong to the Ras homologous (Rho) superfamily of GTP-binding proteins (*Hodge and Ridley, 2016*). They function as binary molecular switches circulating between active (GTP-bound) and inactive (GDP-bound) state. The on-off regulation of the RhoGTPases is under the control of guanine exchange factors (GEFs) and GTPase-activating proteins (GAPs) (*Hodge and Ridley, 2016*). GTP-bound active CDC42 and RAC1 promote the formation of filopodia and lamellipodia, respectively, by activating the NPFs, WASP, and WAVE (*Rohatgi et al., 1999*; *Carlier et al., 1999*). Recent studies revealed a role for kinases in directly activating RhoGTPases mainly through phosphorylation, but the identity of the relevant kinase(s) remained elusive (*Tong et al., 2013*; *Forget et al., 2002*; *Chang et al., 2011*; *Tu et al., 2003*; *Ren et al., 2001*).

ERK3 (MAPK6) is an atypical member of the mitogen-activated protein kinases (MAPKs). Its physiological roles are tissue-specific and can be both dependent and independent of its catalytic activity: kinase-dependent roles of ERK3 include promoting lung cancer cell migration and invasion, while its function in the regulation of breast cancer cell morphology and migration is shown to be partially kinase-independent (*Al-Mahdi et al., 2015*; *Bogucka et al., 2021*; *Bogucka et al., 2020*; *Elkhadragy et al., 2020*). Moreover, ERK3 is a labile protein whose rapid turnover has been implicated in cellular differentiation (*Bogucka et al., 2021*; *Coulombe et al., 2003*). Although the specific signaling mechanisms involved in the regulation of ERK3 have been studied by us and others and provided tissue-specific insights into this atypical MAPK function (*Bogucka et al., 2021*; *Elkhadragy et al., 2020*; *El Merahbi et al., 2020*; *Elkhadragy et al., 2017*; *Long et al., 2012*; *Sauma and Friedman, 1996*), many aspects of ERK3 biology such as its role in suppressing melanoma cell growth and invasiveness remain elusive (*Chen et al., 2019*). It is, however, known that ERK3 possesses a single phosphorylation site at serine 189 (S189) within S-E-G activation motif, located in its kinase domain, which was shown to

be phosphorylated by group I p-21-activated kinases (PAKs), the direct effectors of RAC1 and CDC42 GTPases (*Déléris et al., 2011*; *De la Mota-Peynado et al., 2011*; *Coulombe and Meloche, 2007*).

ERK3 signaling is required for the motility and migration of different cancer cells, yet its direct role in the regulation of the polarized phenotype of the cells and actin cytoskeleton is lacking. Here, we unveil a multilayered role for ERK3 in regulating actin filament assembly. We demonstrate that ERK3 controls bundling of actin filaments into filopodia via activation of CDC42. Mechanistically, ERK3 directly binds to RAC1 and CDC42 in a nucleotide-independent manner and is required for the activity of both RhoGTPases in vivo. Furthermore, ERK3 could function as a direct GEF for CDC42 in vitro. In addition, ERK3 stimulates ARP2/3-dependent actin polymerization and maintains F-actin abundance in cells via direct binding and phosphorylation of the ARP3 subunit at S418. While the kinase activity is not required for CDC42 activation, RAC1 activity partially depends on ERK3 kinase function. The in vivo relevance of these ERK3 functions was corroborated by analyses of the motility of tumor cells in orthotopic mammary tumors grown in mice. Together, our results establish a hitherto unknown regulatory pathway directly controlled by ERK3 involving the major signaling molecules CDC42, RAC1, and ARP2/3 in the modulation of the actin cytoskeleton and cell migration.

## Results

### ERK3 is required for motility of mammary epithelial cells

ERK3 has been shown to regulate mammary epithelial cancer cell migration and metastasis (*Al-Mahdi et al., 2015*; *Bogucka et al., 2020*). To uncover the molecular underpinnings of the role of ERK3 in cell migration, we investigated the intracellular localization of ERK3 by immunocytochemical analyses using a validated antibody (*Bogucka et al., 2021*; *Bogucka et al., 2020*). Endogenous ERK3 co-localized to the F-actin-rich protrusions in human mammary epithelial cells (HMECs) (*Figure 1A*). Reorganization of actin protrusions at the leading edge of the cells initiates cell migration (*Yamaguchi and Condeelis, 2007*; *Affolter and Weijer, 2005*); therefore, we further analyzed the significance of ERK3 in cancer cell motility by loss-of-function studies. Depletion of ERK3 reduced speed and displacement length (distance between the start point and the end position of the cell along each axis) of MDA-MB231 cells (*Figure 1B–F* and *Videos 1–4*). As shown in *Figure 1C*, depletion of ERK3 strongly reduced random cell motility, which remained low over time, as indicated by the calculated acceleration (*Figure 1D*) and concomitantly resulted in a shorter overall displacement length (*Figure 1E*) with shorter migration track length and average speed (*Figure 1—figure supplement 1A and B*). Considering that responses of tumor cells depend on their microenvironment, we tested the in vivo motility of these cells. Intravital imaging of the orthotopic mammary tumors confirmed reduced motility of the ERK3 knockdown MDA-MB231-GFP cells (*Figure 1G* and *Videos 5 and 6*). Together, these data suggest that ERK3 likely controls actin cytoskeleton dynamics, thereby influencing cell shape, motility, and polarized migration.

### ERK3 is activated and required for EGF-mediated directional migration in epithelial cells

To further elucidate the physiological role of ERK3 in breast epithelial cell motility, we assessed EGF-induced chemotactic responses to evaluate directional migratory properties of control and ERK3-depleted primary HMECs and metastatic MDA-MB231 cells. Interestingly, knockdown of ERK3 significantly attenuated the EGF-induced chemotaxis of both cell types (*Figure 1—figure supplement 1C–F*, respectively).

Considering that ERK3 participated in EGF-mediated chemotactic responses of both primary and oncogenic mammary epithelial cells, we tested ERK3 protein kinetics in response to the growth factor. EGF induced serine 189 (S189) phosphorylation of ERK3 at early time points in primary epithelial cells (*Figure 1—figure supplement 2A and B*) and MDA-MB231 cells (*Figure 1—figure supplement 2C and D*). A slight increase in ERK3 protein levels was detected in HMECs upon EGF treatment (*Figure 1—figure supplement 2A and B*). Further cycloheximide (CHX) chase experiments coupled with RT-PCR analyses suggested that EGF treatment did not increase ERK3 protein half-life (*Figure 1—figure supplement 2E–2H*). Taken together, these data confirmed that ERK3 was activated in response to EGF and required for EGF-mediated directional migration of epithelial cells. We then investigated the significance of S189 phosphorylation of ERK3 on F-actin and formation

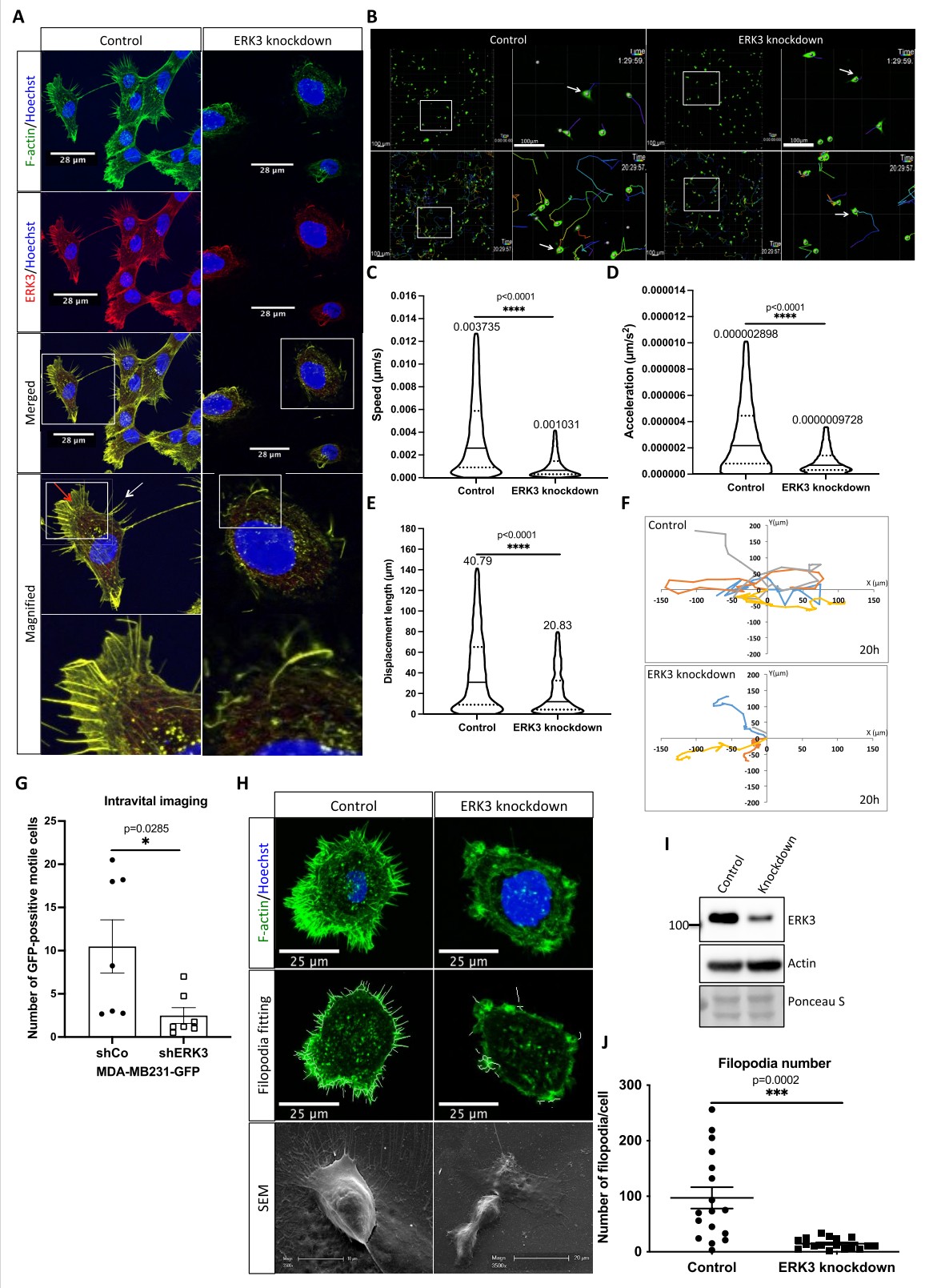

**Figure 1.** ERK3 localizes to the F-actin-rich protrusions and regulates mammary epithelial cells motility. (**A**) Confocal analyses of control (shCo) and *ERK3* knockdown (sh*ERK3*) human mammary epithelial cells (HMECs) co-stained with anti-F-actin (green) and anti-ERK3 antibodies (red). Hoechst staining (blue) was used to visualize cell nuclei. Scale bars 28 µm. Higher magnification images of the boxed areas are shown on the right. Red and white arrows exemplify lamellipodia and filopodia, respectively. (**B–F**) Single-cell tracking analyses of control (shCo) and *ERK3* knockdown (sh*ERK3*) MDA-MB231-

*Figure 1 continued on next page*

*Figure 1 continued*

GFP cells. (**B**) Representative images of the cell migration assay were taken at the start (top panel) and end point (20 hr) (lower panel) of the full track length. Random fields were selected, and boxed areas were magnified on the right. Arrows indicate exemplified cell tracking at the beginning and at the end of the tracking. (**C–E**) Violin plots present cell distribution according to the analyzed motility parameters, calculated as described in the 'Materials and methods' section. Plotted are median (solid line) and 25%/75% quartiles (dashed lines). (**C**) Speed (μm/s), (**D**) acceleration (μm/s$^2$), and (**E**) displacement length (μm) of 14,659 single cells (n = 14659) are depicted. Significance was determined using nonparametric Mann–Whitney test. *p<0.0332, **p<0.0021, ***p<0.0002, ****p<0.0001. Analyses of the migration tracks length and average speed are depicted in *Figure 1—figure supplement 1A and B*. (**F**) Cell migration patterns of four randomly selected control and *ERK3* knockdown cells were plotted as x–y trajectories over the 20 hr tracking. (**G**) Quantification of the intravital imaging of the control (shCo) and *ERK3* knockdown (sh*ERK3*) orthotopic mammary tumors (MDA-MB231-GFP). The number of GFP-positive motile cells was quantified as described in the 'Materials and methods' section and is presented as mean ± SEM from seven animals (n = 7) per condition. Please see representative *Videos 5 and 6*. Chemotactic responses of the *ERK3*-depleted cells were assessed in the presence of EGF and are presented in *Figure 1—figure supplement 1C–F*. ERK3 protein kinetics and stability in response to EGF are depicted in *Figure 1—figure supplement 2*. (**H–J**) Depletion of *ERK3* alters leading edge protrusions in primary mammary epithelial cells. (**H**) Confocal F-actin (green)/Hoechst (blue) and SEM images analyses of control and *ERK3* knockdown HMECs are presented after 4 hr starvation. For the confocal analyses, exemplified filopodia fitting was included in the middle panel. (**I**) Western blot analyses presenting *ERK3* knockdown efficiency and total levels of β-actin. Ponceau S was used as a loading control. (**J**) Filopodia number was quantified as described in the 'Materials and methods' section and is presented as mean ± SEM of 17 single cells (n = 17); *p<0.0332, **p<0.0021, ***p<0.0002, ****p<0.0001, unpaired *t*-test. Effect of S189 phosphorylation of ERK3 on F-actin assembly is presented in *Figure 1—figure supplement 3*.

The online version of this article includes the following source data and figure supplement(s) for figure 1:

**Source data 1.** Prism and Excel file for *Figure 1C–E*.

**Source data 2.** Excel file for *Figure 1F*.

**Source data 3.** Prism and Excel file for *Figure 1G*.

**Source data 4.** Full membrane scans for western blot images for *Figure 1I*.

**Source data 5.** Prism and Excel file for *Figure 1J*.

**Figure supplement 1.** ERK3 regulates mammary epithelial cells motility.

**Figure supplement 1—source data 1.** Prisma file and original movie files for *Figure 1—figure supplement 1A and B*.

**Figure supplement 1—source data 2.** Full membrane scans for western blot images for *Figure 1—figure supplement 1C and E*.

**Figure supplement 1—source data 3.** Prism and Excel file for *Figure 1—figure supplement 1D*.

**Figure supplement 1—source data 4.** Prism and Excel file for *Figure 1—figure supplement 1F*.

**Figure supplement 2.** Effect of EGF on chemotactic responses and ERK3 protein kinetics in human mammary epithelial cells (HMECs).

**Figure supplement 2—source data 1.** Full membrane scans for western blot images for *Figure 1—figure supplement 2A, C, and E*.

**Figure supplement 2—source data 2.** Prism and Excel file for *Figure 1—figure supplement 2B*.

**Figure supplement 2—source data 3.** Prism and Excel file for *Figure 1—figure supplement 2D*.

**Figure supplement 2—source data 4.** Prism and Excel file for *Figure 1—figure supplement 2F and G*.

**Figure supplement 2—source data 5.** Prism and Excel file for *Figure 1—figure supplement 2H*.

**Figure supplement 3.** Effect of overexpressed S189 mutants on F-actin.

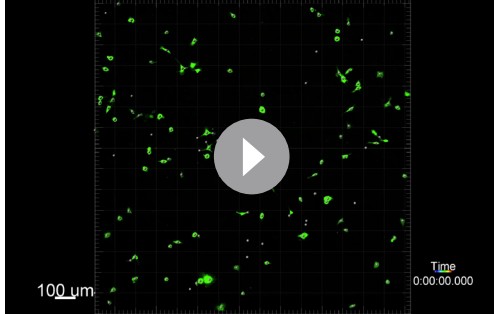

**Video 1.** Movies representing motility of control (*Videos 1 and 2*) and ERK3-knockdown (*Videos 3 and 4*) MDA-MB231 cells.

https://elifesciences.org/articles/85167/figures#video1

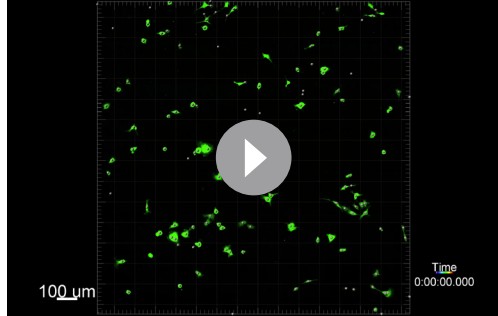

**Video 2.** Movies representing motility of control (*Videos 1 and 2*) and *ERK3*-knockdown (*Videos 3 and 4*) MDA-MB231 cells.

https://elifesciences.org/articles/85167/figures#video2

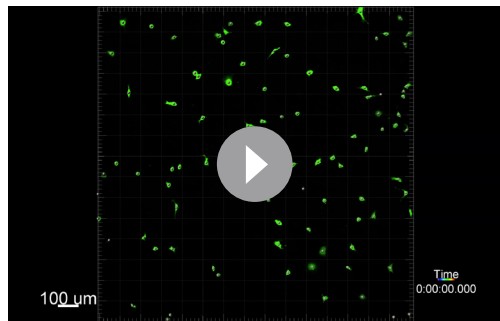

**Video 3.** Movies representing motility of control (*Videos 1 and 2*) and *ERK3*-knockdown (*Videos 3 and 4*) MDA-MB231 cells.

https://elifesciences.org/articles/85167/figures#video3

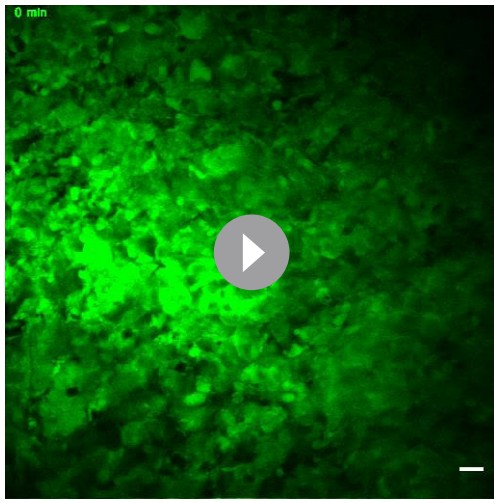

**Video 5.** Representative videos following the motility of the control (shCo) and *ERK3* knockdown (shERK3) *GFP*-expressing MDA-MB231 cells.

https://elifesciences.org/articles/85167/figures#video5

of actin-rich protrusions. ERK3-depleted HMECs cells (shERK3 – 3′ UTR) presented in *Figure 1A* were transfected with V5-tagged ERK3 wild type (ERK3 WT) or S189 mutants (ERK3 S189A/ERK3 S189D) and expression of exogenous ERK3 was visualized by V5-tag staining. Co-staining of the cells with phalloidin green did not show any significant differences in F-actin organization upon the overexpression of the S189 phosphorylation mutants compared to the wild type ERK3 transfected cells (*Figure 1—figure supplement 3*). Moreover, we observed that exogenous ERK3 localized predominantly in the cytosol compared to the endogenous ERK3, which we detected at the cell edges and F-actin rich protrusions (*Figure 1A*). These results prompted us to prefer ERK3 knockdown cells for the study of ERK3-mediated phenotypes.

## ERK3 is required for actin-rich protrusions

In order to migrate, cells acquire morphological asymmetry to drive their locomotion. One of the first steps into breaking spatial symmetry of the cell involves its polarization and formation of membrane extensions (*Theriot and Mitchison, 1991*). In response to extracellular guidance, cells polarize and extend F-actin-rich protrusions at the leading edge to direct cell migration (*Affolter and Weijer, 2005*; *Ridley et al., 2003*; *Lauffenburger and Horwitz, 1996*). As shown in *Figure 1A*, ERK3 co-localized with F-actin at the edge of the cell and at the protruding filopodial spikes. Moreover, knockdown of ERK3 clearly reduced F-actin content and actin-rich protrusions in human mammary primary

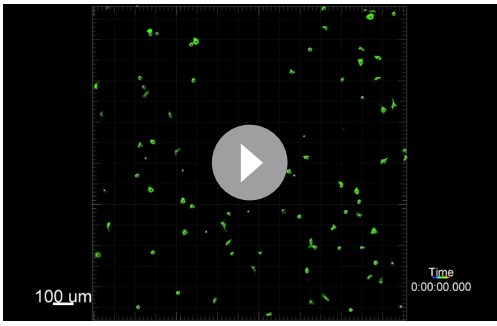

**Video 4.** Movies representing motility of control (*Videos 1 and 2*) and *ERK3*-knockdown (*Videos 3 and 4*) MDA-MB231 cells.

https://elifesciences.org/articles/85167/figures#video4

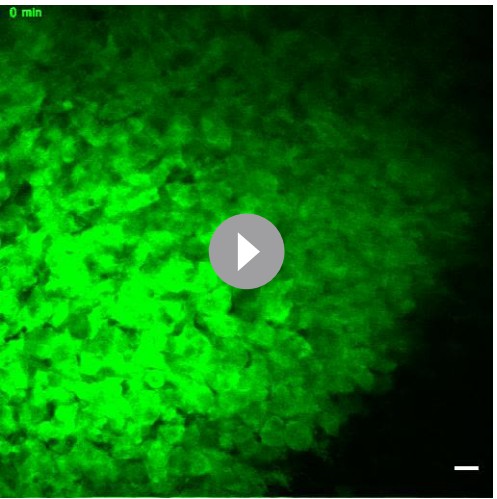

**Video 6.** Representative videos following the motility of the control (shCo) and *ERK3* knockdown (shERK3) *GFP*-expressing MDA-MB231 cells.

https://elifesciences.org/articles/85167/figures#video6

epithelial cells (*Figure 1A*). Further analyses of control and ERK3 knockdown HMECs at the single-cell level confirmed that ERK3-depleted cells had limited, diffuse F-actin, concentrated in the cortical areas of the cells, and a significantly decreased number of filopodia (*Figure 1G–J*).

## ERK3-dependent regulation of RAC1 and CDC42 activity

RhoGTPases are the major membrane signaling transmitters that drive polarized cell asymmetry by locally regulating the formation of F-actin-rich protrusions (*Hall, 1998*; *Nobes and Hall, 1995*). EGF signaling triggers activation of RAC1 and CDC42, which link the signal to the actin cytoskeleton leading to the formation of lamellipodia- and filopodia-like protrusions, respectively (*Figure 2A*; *Kurokawa et al., 2004*).

Considering that loss of ERK3 significantly affected F-actin distribution and filopodia formation in primary mammary epithelial cells (*Figure 1A and G–J*), we tested the activity of the two major regulators of actin assembly, CDC42 and RAC1 (*Figure 2B–G*). ERK3 knockdown significantly decreased the levels of both basal and EGF-induced GTP-bound CDC42 and RAC1 in primary (HMEC) (*Figure 2B, D, and F*) and triple-negative breast cancer (MDA-MB231) mammary epithelial cells (*Figure 2C, E, and G*), respectively. HMECs were cultured in growth factor/hormone-enriched medium that also contained EGF. Therefore, we withdrew the supplements to test the effect of EGF alone on the activity of RAC1 and CDC42 in these cells and compared the responses with those of the cancer cells. Interestingly, we detected high RAC1 and CDC42 levels at steady state in HMEC cells, which underwent a rapid activation-inactivation cycle after starvation and restimulation with EGF alone. The inhibitory effect of ERK3 knockdown on RAC1 was not as striking as with CDC42 (*Figure 2D–G*). Intriguingly, we could observe co-precipitation of ERK3 protein with the active RhoGTPases at endogenous levels, suggesting that these proteins exist in a multimeric complex (*Figure 2B and C*). We expanded these analyses to different cell types and consistently found that depletion of ERK3 reduced the activity of RAC1 and CDC42 and that the active RhoGTPases specifically co-precipitated endogenous ERK3 (*Figure 2—figure supplements 1 and 2*). Interestingly, the effect of ERK3 on RAC1 and CDC42 activity is either EGF-dependent (MCF7 cells [*Figure 2—figure supplement 1A–C*] and Calu1 cells [*Figure 2—figure supplement 2A–C*]), EGF-independent (MDA-MB468 cells [*Figure 2—figure supplement 1D–F*]) or occurred both in resting and EGF-treated cells (HT-29 [*Figure 2—figure supplement 2D–F*]). Apart from cancer cells, we were also able to confirm that ERK3 sustains the activity of RAC1 and CDC42 in primary human umbilical vein endothelial cells, HUVECs (*Figure 2—figure supplement 2G–I*).

## ERK3 directly binds to RAC1 and CDC42

In light of these observations, we tested whether any direct interaction existed between these RhoGTPases and ERK3. By employing purified full-length recombinant proteins, we found that ERK3 directly bound to RAC1 and CDC42 in a nucleotide-independent manner (*Figure 2H and I*). The nucleotide-loading status of the purified RAC1 and CDC42 proteins was simultaneously assessed by GST-PAK1-PBD pull-down assay followed by immunoblots (*Figure 2—figure supplement 3*). To further assess the affinity of the binding, we performed concentration-dependent protein binding assays with ELISA as described in the 'Materials and methods' section. Binding of full-length ERK3 to RAC1 and CDC42 could be detected at a low nanomolar range (5 nM) (*Figure 2J and K*). Next, we tested whether the kinase domain of ERK3 (amino acids [aa] 9–327) would be sufficient for this interaction. In vitro GST-pull-down assays confirmed that the kinase domain of ERK3 was sufficient for its binding to RAC1 and CDC42 (*Figure 2L* and *Figure 2—figure supplement 4*).

## ERK3 functions as a GEF for CDC42

The intrinsic GDP-GTP exchange of RhoGTPases is a slow process, which is stimulated in vivo by GEFs (*Figure 3A*; *Hodge and Ridley, 2016*). To examine the ability of the ERK3 kinase domain to stimulate GDP-GTP exchange, we incubated GDP-loaded RhoGTPases with non-hydrolysable GTPγS in the presence or absence of the ERK3 kinase domain and quantified the final amount of GTP-bound CDC42 (*Figure 3B and C*) and RAC1 (*Figure 3D and E*) using GST-PAK1-PBD fusion beads. The ERK3 kinase domain failed to interact directly with the GST-PAK1-PBD protein but bound to active CDC42 in the same assay confirming that conformationally stable protein was employed in these assays (*Figure 2—figure supplement 4*). These experiments showed that although ERK3 regulated the activity of both

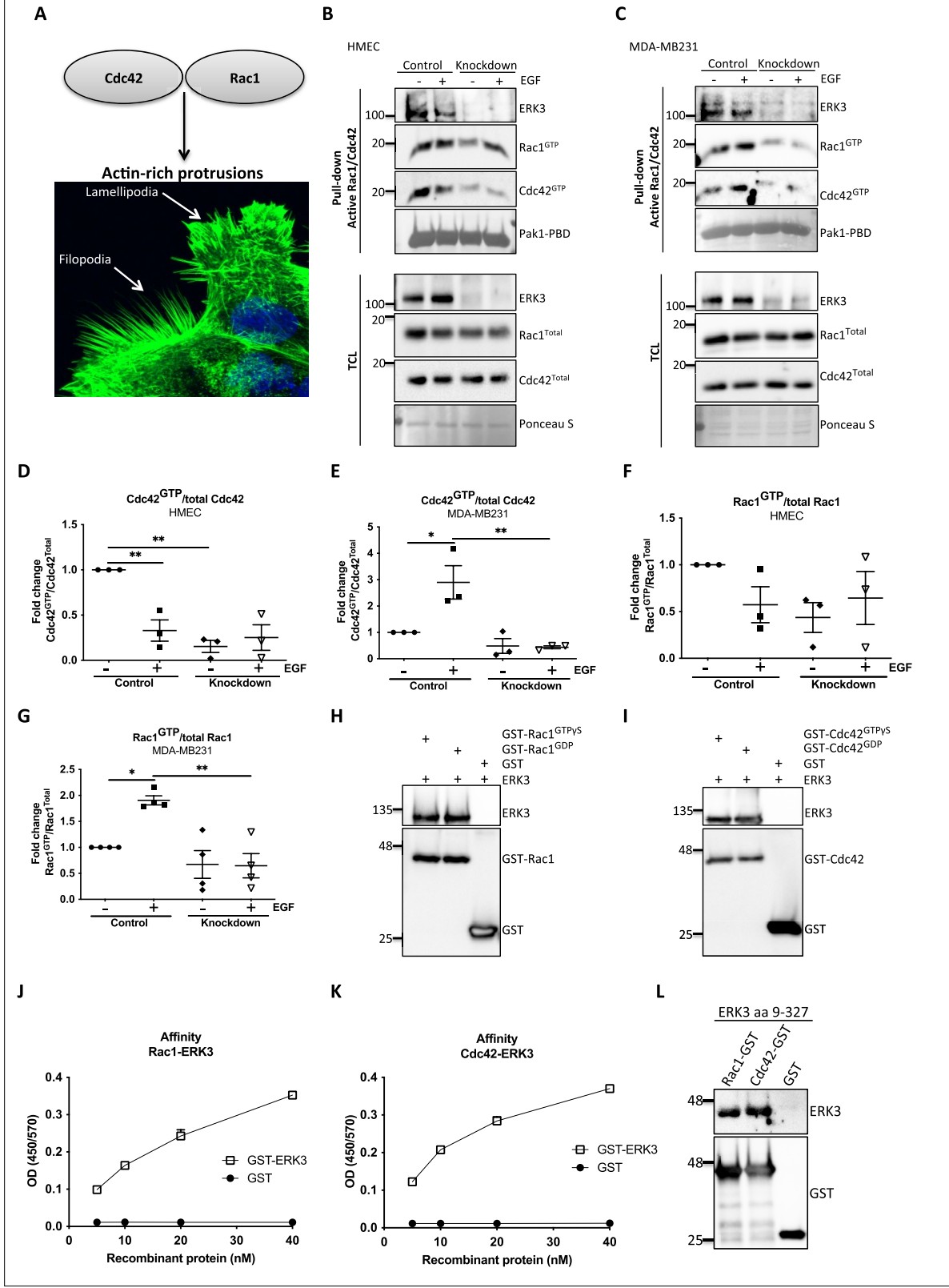

**Figure 2.** ERK3 binds RAC1 and CDC42 and regulates their activity. (**A**) Confocal image of phalloidin-stained actin-rich protrusions in human mammary epithelial cells (HMECs), filopodia and lamellipodia. (**B–G**) In vivo pull-down of active RAC1 and CDC42 from control and *ERK3*-depleted (knockdown) (**B**) HMECs and (**C**) MDA-MB231 cells in the presence and absence of EGF. Levels of active (GTP-bound) CDC42 and RAC1 as well as the total protein expression were assessed and quantified. ERK3 knockdown efficiency was verified in the total cell lysate (TCL) as well as the co-immunoprecipitation of

*Figure 2 continued on next page*

*Figure 2 continued*

ERK3 with active RAC1 and CDC42. Relative levels of (**D, E**) active CDC42 and (**F, G**) RAC1 were calculated with respect to the total protein levels and are presented as mean ± SEM from minimum three (n = 3) independent experiments; *p<0.0332, **p<0.0021, ***p<0.0002, ****p<0.0001, one-way ANOVA, Tukey's post-test. ERK3-dependent regulation of the RAC1 and CDC42 activity was assessed in multiple cell types and data are presented in *Figure 2—figure supplements 1 and 2*. (**H, I**) In vitro interaction between ERK3 and GDP/GTP bound (**H**) RAC1 and (**I**) CDC42 was assessed, GST was used as a negative control. Pull-down efficiency was assessed with GST antibody and levels of bound ERK3 were verified. The nucleotide-loading status of the purified RAC1 and CDC42 proteins is presented in *Figure 2—figure supplement 3*. (**J, K**) Concentration-dependent binding affinity of ERK3-GST protein to (**J**) RAC1 and (**K**) CDC42 was determined. Interacting ERK3/GST proteins were used at 5, 10, 20, and 40 nM concentrations. Data from three independent experiments run in triplicates are presented as mean ± SEM. (**L**) ERK3 kinase domain (aa 9–327) binds to RAC1 and CDC42. Representative analysis of the in vitro GST pull-down of RAC1 and CDC42 and the interaction with the ERK3 kinase domain recombinant protein. Binding affinity of ERK3 (aa 9–327) with PAK1-PBD was verified and is presented in *Figure 2—figure supplement 4*.

The online version of this article includes the following source data and figure supplement(s) for figure 2:

**Source data 1.** Full membrane scans for western blot images for *Figure 2B, C, H, I, and L*.

**Source data 2.** Prism and Excel file for *Figure 2D*.

**Source data 3.** Prism and Excel file for *Figure 2E*.

**Source data 4.** Prism and Excel file for *Figure 2F*.

**Source data 5.** Prism and Excel file for *Figure 2G*.

**Source data 6.** Prism and Excel file for *Figure 2J*.

**Source data 7.** Prism and Excel file for *Figure 2K*.

**Figure supplement 1.** Pull-down of active RAC1 and CDC42 from control and *ERK3*-depleted (si*ERK3* knockdown) breast cancer cells.

**Figure supplement 1—source data 1.** Full membrane scans for western blot images for *Figure 2—figure supplement 1A and D*.

**Figure supplement 1—source data 2.** Prism and Excel file for *Figure 1—figure supplement 1B and C*.

**Figure supplement 1—source data 3.** Prism and Excel file for *Figure 1—figure supplement 1E and F*.

**Figure supplement 2.** ERK3 binds Rac1 and Cdc42 and regulates their activity.

**Figure supplement 2—source data 1.** Full membrane scans for western blot images for *Figure 2—figure supplement 2A, D, and G*.

**Figure supplement 2—source data 2.** Prism and Excel file for *Figure 2—figure supplement 2A and C*.

**Figure supplement 2—source data 3.** Prism and Excel file for *Figure 2—figure supplement 2E and F*.

**Figure supplement 2—source data 4.** Prism and Excel file for *Figure 2—figure supplement 2H and I*.

**Figure supplement 3.** In vitro pull-down of purified native and GTP/GTP-loaded RAC1 and CDC42 proteins using PAK1-PBD-fusion beads as described in the 'Materials and methods' section.

**Figure supplement 3—source data 1.** Full membrane scans for western blot images for *Figure 2—figure supplement 3*.

**Figure supplement 4.** Western blot analyses of in vitro pull-down assay assessing binding affinity of ERK3 (aa 9–327) and PAK1-PBD.

**Figure supplement 4—source data 1.** Full membrane scans for western blot images for *Figure 2—figure supplement 4*.

---

CDC42 and RAC1 in cultured cells, it only directly stimulated the GDP-GTP exchange of CDC42 (*Figure 3B and C*). We further corroborated these results by measuring the GEF activity of the ERK3 kinase domain in vitro by employing a second fluorophore-based assay. The results confirmed those of the previous assay as addition of ERK3 stimulated GTP-loading on CDC42 (*Figure 3F*) but not on RAC1 (*Figure 3G*). Notably, ERK3 exerted a more potent and long-lasting effect than Dbl's Big Sister (Dbs), a well-established GEF for CDC42 and RhoA proteins (*Figure 3F*; *Baumeister et al., 2006*).

To assess whether full-length ERK3 could potentiate the effect in terms of activation of CDC42, we compared GEF activity of the ERK3 kinase domain with that of the full-length protein. Interestingly, full-length ERK3 exhibited weaker GEF activity in vitro than the kinase domain of ERK3 (*Figure 3—figure supplement 1A*). The observed in vitro discrepancy between the kinase domain and the full-length protein could be attributed to differences in posttranslational modifications due to different production methods of the two recombinant proteins, which might affect conformation and/or activity. The full-length protein employed in these assays was purified from Sf9 cells, while the kinase domain was purified from bacterial cells and therefore lacked – among other things – Ser189 phosphorylation and thus kinase activity in vitro (*Figure 3—figure supplement 1B*). We next examined whether ERK3 is required for the localization and activity of CDC42 at the plasma membrane (PM). Cellular fractionation assays revealed that knockdown of ERK3 does not disrupt the localization of CDC42 and RAC1 to the PM (*Figure 4A–C*). On the contrary, the PM fraction from ERK3-depleted HMECs had more

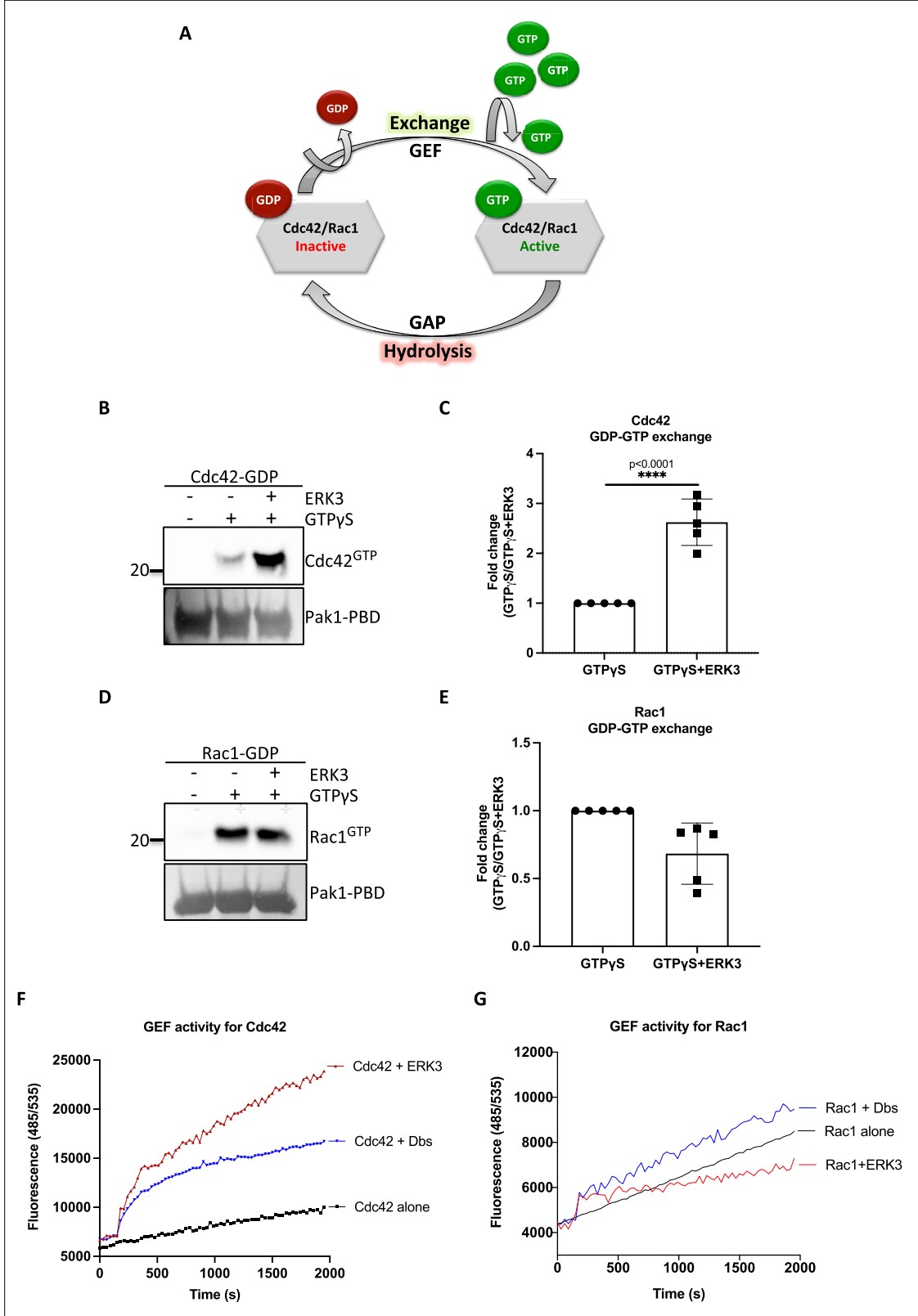

**Figure 3.** ERK3 kinase domain binds to RAC1 and CDC42 and facilitates nucleotide exchange on CDC42. (**A**) Schematic representation of the GDP-GTP nucleotide exchange on CDC42 and RAC1. Guanine exchange factor (GEF) facilitates the GDP-GTP exchange on the RhoGTPases, resulting in the GTP-bound form of the CDC42 or RAC1. GTPase-activating protein (GAP) stimulates GTP hydrolysis. (**B–E**) Role of ERK3 in the GDP-GTP exchange was assessed in the in vitro assays as described in the 'Materials and methods' section for (**B, C**) CDC42 and (**D–, E**) RAC1. (**B, D**) Representative western

*Figure 3 continued on next page*

*Figure 3 continued*

blot analyses of active RAC1 and CDC42 pull-down, respectively, using PAK1-PBD fusion beads. Levels of active RhoGTPases were detected using RAC1- and CDC42-specific antibodies. Levels of PAK1-PBD protein were detected by Ponceau S staining and used for the quantification presented in (**C**) and (**E**). Fold change in GTP-loading was calculated by normalization of the signal obtained for the samples with GTPγS in the presence of ERK3 with samples incubated with GTPγS alone and is presented as mean ± SEM from (**D**) five (n = 5) independent experiments; *p<0.0332, **p<0.0021, ***p<0.0002, ****p<0.0001, unpaired *t*-test. (**F, G**) In vitro RhoGEF activity assay was performed to assess guanine nucleotide exchange activity on (**F**) CDC42 and (**G**) RAC1 in the presence and absence of recombinant ERK3 protein (9–327 aa). After six initial readings, Dbs-GEF or ERK3 protein were added at 0.5 μM final concentration. GEF activity was expressed as mean relative fluorescence units (RFUs) from at least three independent experiments. RhoGEF Dbs protein was used as a positive control. In vitro GEF activity of full-length ERK3 vs. ERK3 kinase domain towards CDC42 was compared as is presented in *Figure 3—figure supplement 1A*.

The online version of this article includes the following source data and figure supplement(s) for figure 3:

**Source data 1.** Full membrane scans for western blot images for *Figure 3B and D*.

**Source data 2.** Prism and Excel file for *Figure 3C*.

**Source data 3.** Prism and Excel file for *Figure 3E*.

**Source data 4.** Prism and Excel file for *Figure 3F*.

**Source data 5.** Prism and Excel file for *Figure 3G*.

**Figure supplement 1.** The ERK3 kinase domain regulates GEF activity of Cdc42.

**Figure supplement 1—source data 1.** Prism and Excel file for *Figure 3—figure supplement 1*.

**Figure supplement 1—source data 2.** Full membrane scans for the western blot images for *Figure 3—figure supplement 1B*.

---

total CDC42 and RAC1 than control cells. Interestingly, subsequent pull-down of active CDC42/RAC1 from the PM fraction revealed that although both RhoGTPases are present at the plasma membrane in the absence of ERK3, their activity is significantly reduced (*Figure 4B and C*). These results were corroborated by the colocalization of ERK3 with CDC42 to the protrusions at the leading edge of the cell (*Figure 4D–G*) with mean Pearson's correlation coefficient (PCC (r)) of 0.6638 ± 0.02946 and Spearman's rank correlation coefficient (SRCC ($\rho$)): 0.7122 ± 0.02586 (*Figure 4F and G*, respectively).

## ERK3 as a nucleation-promoting factor of ARP2/3-dependent actin polymerization

Active RAC1 and CDC42 induce ARP2/3-dependent actin polymerization by activating their effector NPFs, WASP and WAVE, respectively (*Rohatgi et al., 1999*; *Carlier et al., 1999*). *Figure 5A* depicts a well-described signaling module of ARP2/3-dependent actin nucleation stimulated by CDC42-activated WASP protein (*Rohatgi et al., 1999*; *Espinoza-Sanchez et al., 2018*; *Padrick et al., 2011*; *Insall and Machesky, 2009*; *Kelly et al., 2006*).

As ERK3 protein co-precipitated with active RAC1 and CDC42 (*Figure 2B and C*), we further investigated whether also components of the ARP2/3 complex precipitate with RAC1/CDC42. Interestingly, we readily detected the ARP2/3 complex subunits ARP3, ARP2, and ARPC1A, as well as ERK3 by immunoblots in active RAC1/CDC42 pull-downs (*Figure 5B*). Consistent with the reduction in active RAC1 and CDC42 levels, we observed that knockdown of ERK3 by CRISPR/Cas9 or with si/shRNAs led to a reduction in F-actin staining in mammary epithelial cells as shown in *Figure 5C* and *Figure 5—figure supplement 1A*. To further quantify these effects and corroborate the role of ERK3 in polymerization of actin filaments, we evaluated the ratio of cytoskeleton-incorporated filamentous (F-) and globular/monomeric actin (G-actin) in control and ERK3-depleted mammary epithelial cells by ultracentrifugation. Knockdown of ERK3 significantly shifted total F-actin to G-actin in both primary HMECs (*Figure 5D and E*) and MDA-MB231 cancer cells (*Figure 5—figure supplement 1B and C*). Moreover, in the absence of CDC42, we could still detect colocalization between endogenous ERK3 and the ARP3 subunit of the ARP2/3 complex (*Figure 5—figure supplement 2A–D*). These data prompted us to test whether ERK3 directly controls ARP2/3-dependent actin polymerization in the absence of RAC1 and CDC42.

In vitro, the ARP2/3 complex expresses low intrinsic actin nucleating activity. In vivo, its activity is tightly controlled by NPFs, which stimulate conformational changes (*Mullins et al., 1998*; *Welch et al., 1998*). To determine whether ERK3 had any direct effect on the assembly of actin filaments, we used a pyrene-actin polymerization assay to monitor fluorescence kinetics over time as the actin filaments assembled in the presence of the purified ARP2/3 complex. We used the VCA domain

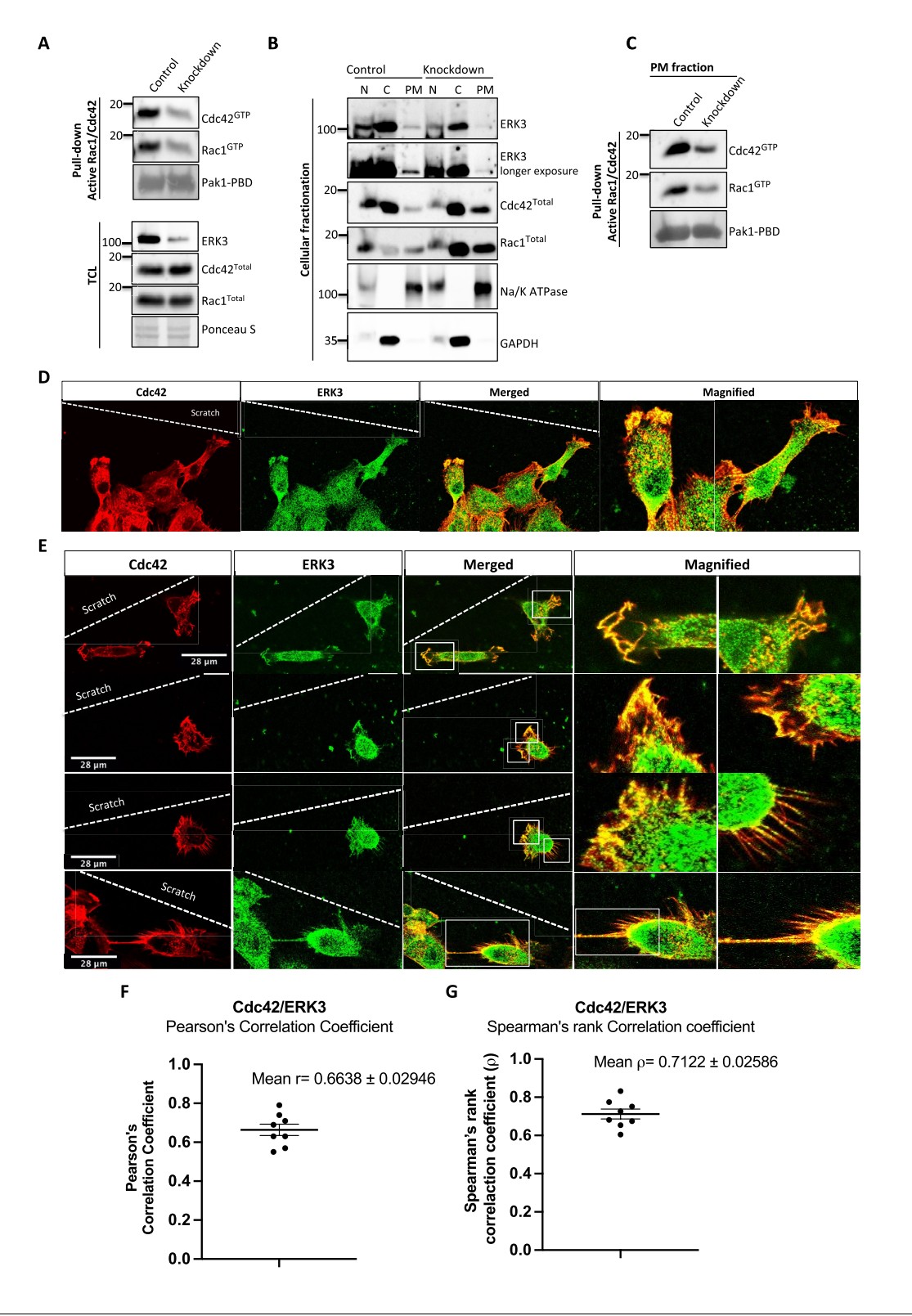

**Figure 4.** ERK3 colocalizes with CDC42 in polarized cells and regulates activity of both CDC42 and RAC1 at the plasma membrane (PM). (**A–C**) Cell fractionation for control (siCo) and *ERK3*-depleted HMECs (si*ERK3*). Fractionation was performed using Minute Plasma Membrane Protein Isolation and Cell Fractionation Kit (Cat# SM-005, Invent Biotechnologies) according to the manufacturer's instruction. (**A**) Total cell lysate control was used to determine knockdown efficiency and activation status of RAC1 and CDC42 using active RAC1/CDC42 pull-down according to the manufacturer's

*Figure 4 continued*

instructions (Cat# 16118/19, Thermo Fisher) and as described in the 'Materials and methods' section. (**B**) Cellular fractionation was performed. Expression levels of *ERK3*, *RAC1*, and *CDC42* were assessed in nuclear (N), cytosolic (C), and PM fractions. Na/K ATPases and GAPDH were used as controls for the analyzed fractions. (**C**) Active RAC1/CDC42 pull-down was performed from the isolated PM fraction. (**D–G**) Colocalization of ERK3 and CDC42 in polarized cells. Human mammary epithelial cells (HMECs) were seeded and cultured on cover slips. When cells became around 70% confluent, scratch wounds were introduced to the cover slip using a 200 µl tip, medium was exchanged to supplement-free, and cells were cultured for additional 6 hr. Afterward, cells were fixed and subjected to IF staining as described in the 'Materials and methods' section with anti-CDC42 (red) (secondary antibody: anti-mouse Cy3; Cat# A10521, Thermo Fisher Scientific) and anti-ERK3 (green) (secondary antibody: Alexa Fluor 488; Cat# A11008, Thermo Fisher Scientific) antibodies. Merged images show the colocalization of both proteins in yellow. Magnification of the boxed regions is shown on the right for better visualization. In (**D**), a group of cells at the scratch site is presented and ERK3-CDC42 colocalization is marked with red arrows at the cell protrusions and with white arrows at the cell body. (**E**) Images representing ERK3-CDC42 colocalization at a single-cell level at the scratch site. Scale bars 28 µm. (**F, G**) Colocalization of ERK3 and CDC42 was analyzed as described in the 'Materials and methods' section and values for (**F**) Pearson's correlation coefficient as well as the (**G**) Spearman's rank correlation coefficient are presented for eight randomly selected cells (n = 8). Scores above 0 indicate a tendency towards colocalization with a perfect colocalization with a score of 1.

The online version of this article includes the following source data for figure 4:

**Source data 1.** Full membrane scans for western blot images for *Figure 4A–C*.

**Source data 2.** Prism and Excel file for *Figure 4F and G*.

of WASP protein as a positive control. Indeed, recombinant ERK3 stimulated ARP2/3-dependent actin polymerization at a nanomolar concentration (5 nM) in the presence of 10 nM of the ARP2/3 complex (*Figure 5F*). Moreover, the nucleation-promoting activity of ERK3 was comparable to WASP (VCA), a well-known NPF. ERK3 protein alone did not exert any stimulatory effect on actin nucleation (*Figure 5F*). These data suggest that ERK3 could function as nucleation promoting factor that stimulates ARP2/3-dependent actin polymerization. Additionally, we measured the effect of having both ERK3 and WASP (VCA) present on ARP2/3-dependent actin polymerization. Using the same concentration as for the initial screening (ERK3 4.8 nM and WASP [VCA] at 400 nM), we did not detect any additive effect when both proteins were combined (*Figure 5—figure supplement 3*). Combination of both proteins under this stoichiometric setting rather negatively affected the polymerization rate that could be achieved by each protein separately (*Figure 5—figure supplement 3*).

## ERK3 directly binds to ARP3

The ARP2/3 complex is composed of seven subunits: the actin-related proteins ARP2 and ARP3 and five associated proteins (ARPC1-5) that sequester ARP2 and ARP3 in the complex's inactive conformation (*Robinson et al., 2001*; *Nolen et al., 2004*; *Figure 6A*). In vitro, NPFs such as WASP stimulate actin polymerization by directly binding to and activating ARP2/3. Considering that ERK3 exerted a similar effect on the ARP2/3-dependent actin polymerization as NPFs, we further investigated the mode of interaction between ERK3 and the ARP2/3 protein complex by employing purified components. Indeed, using quantitative ELISAs we found that full-length ERK3 bound to the ARP2/3 complex with high-affinity in vitro (*Figure 6B*).

ARP2 and ARP3 are essential for the nucleating activity of the ARP2/3 complex and occupy the central position in actin nucleation. Structural modeling has suggested that although ARP2 and ARP3 interact with each other in the inactive conformation, they are not in the right configuration to act as a seed for actin nucleation. After activation by NPF binding, the heterodimer mimics two actin subunits. Binding of an actin monomer then initiates the formation of the actin nucleus and the polymerization process (*Mullins et al., 1998*; *Gournier et al., 2001*; *Kelleher et al., 1995*). Extensive characterization of the individual subunits also revealed that ARP3 is crucial for the stimulation and activity of the ARP2/3 complex and nucleation in vitro (*Gournier et al., 2001*). Considering that ARP3 is a structural component regulating the nucleating properties of the ARP2/3 complex, we further determined whether ERK3 can directly bind to ARP3. Through GST pull-down experiments with purified recombinant proteins, we could indeed detect a concentration-dependent direct binding between full-length ERK3 and ARP3 (*Figure 6C*). These results were further confirmed with ELISA analyses suggesting a high-affinity binding between these two proteins (*Figure 6D*). The ERK3 kinase domain binds also directly to the Arp3 protein, albeit with low affinity in comparison to the full-length protein (*Figure 6-figure supplement 1A*). Finally, we also detected ERK3 co-precipitating with ARP3 and ARP2 in HMECs at endogenous levels (*Figure 6E*).

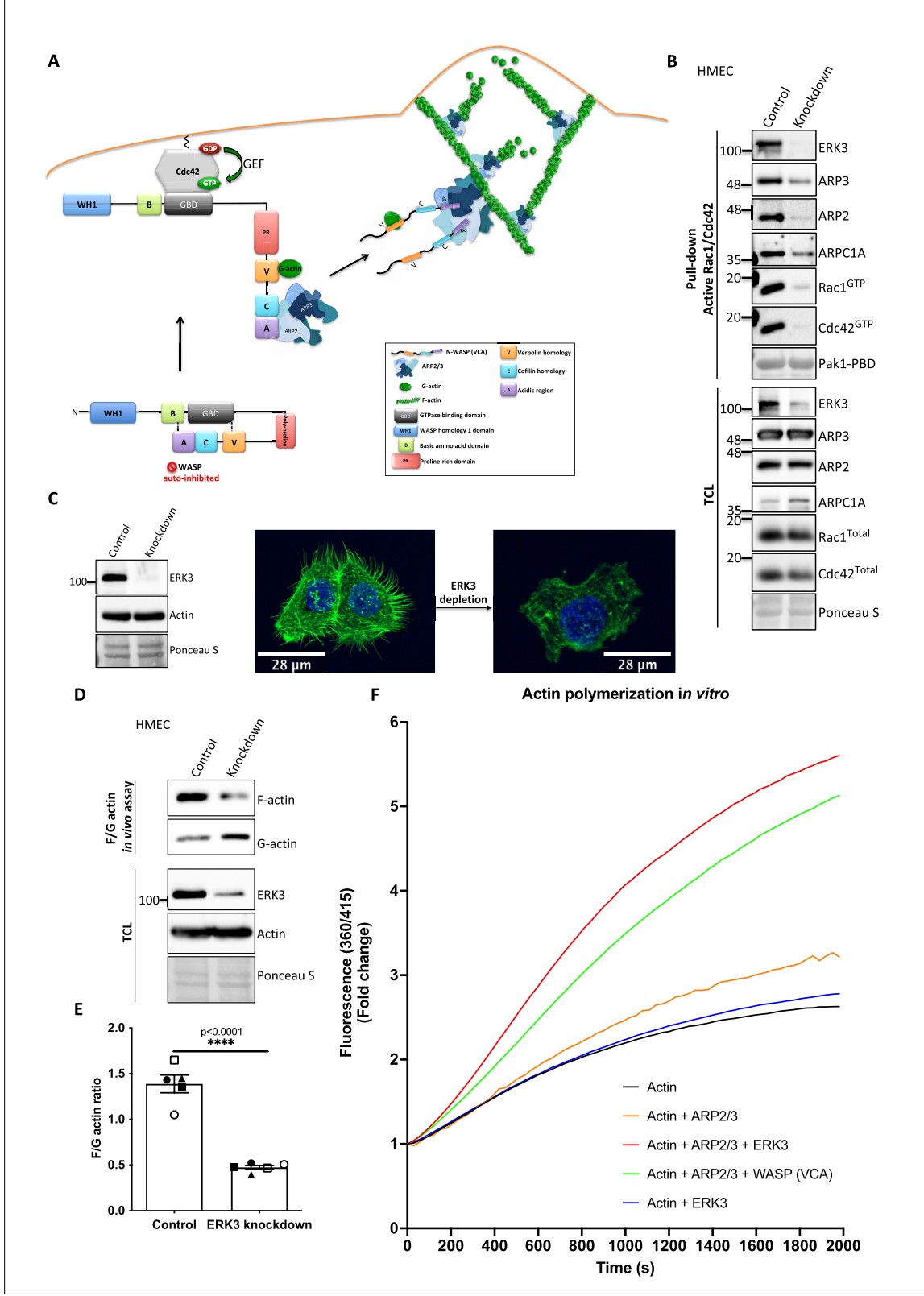

**Figure 5.** ERK3 acts as a nucleation-promoting factor (NPF) in ARP2/3-dependent actin polymerization in vitro and in vivo. (**A**) Schematic overview of CDC42-WASP-stimulated ARP2/3-dependent actin polymerization based on the cited literature. The process involves ARP2/3 complex, WASP (VCA) as nucleation promoting factor, filamentous actin (F-actin), and monomeric actin (G-actin). In the initial step, CDC42 is activated by GEF-catalyzed exchange of GDP to GTP. Active CDC42 (CDC42-GTP) binds to the GTP-binding domain (GBD) on WASP, thereby displacing the VCA domain. While

*Figure 5 continued on next page*

*Figure 5 continued*

the V-verpolin-like motif binds actin monomer (G-actin), C-central and A-acidic domains bind and activate the ARP2/3 complex. Conformational changes induced by the binding of the ARP2/3 complex promote its binding to the actin filament, which is strengthened by the additional interaction of the ARP2/3 complex with WASP (VCA)-G-actin. Further conformational changes will secure the ARP2/3 complex on the filament and allow its binding to the actin monomer and the polymerization of the newly nucleated filament. Actin polymerizes at the fast-growing/barbed end, elongating toward the plasma membrane and the ARP2/3 complex would cross-link newly polymerizing filament to the existing filament. (**B**) ERK3 co-precipitates with active RAC1 and CDC42 in complex with ARP2/3. Active RAC1/CDC42 pull-down was performed using control and ERK3 knockdown human mammary epithelial cells (HMECs). Levels of the active RAC1 and CDC42 were assessed as well as the co-immunoprecipitation levels of ERK3, ARP2, ARP3, and ARPC1A. Levels of the total protein expression were evaluated in the total cell lysates (TCL) and Ponceau S staining was used as a loading control. (**C–F**) ERK3 regulates F-actin levels in vitro and in vivo. (**C**) Western blot analyses of control (CRISPR Co) and *ERK3*-depleted (CRISPR *ERK3*) HMECs are presented alongside with representative confocal images of F-actin staining. (**D, E**) In vivo analysis of F- and G-actin levels in HMECs upon *ERK3* knockdown. (**D**) Representative western blot analyses of the enriched F- and G-actin fractions as well as the ERK3 knockdown validation and total actin levels in the TCL are presented. (**E**) F- and G-actin levels were quantified, and ratios were calculated from five (n = 5) independent experiments and are presented as mean ± SEM; *p<0.0332, **p<0.0021, ***p<0.0002, ****p<0.0001, unpaired *t*-test. Analyses of ERK3-dependent regulation of F-actin levels in cancerous MDA-MB231 cells is presented in *Figure 5—figure supplement 1*. Cellular colocalization between endogenous ERK3 and the ARP2/3 was assessed in the absence of CDC42 and is presented in *Figure 5—figure supplement 2*. (**F**) Effect of full-length ERK3 on ARP2/3-dependent pyrene actin polymerization was assessed using a pyrene actin polymerization assay. Polymerization induced by the VCA domain of WASP that served as a positive control (green) as well as the ARP2/3 (orange) and ERK3 protein alone (blue) are shown for reference. Actin alone (black) was used to establish a baseline of polymerization. Fluorescence at 360/415 was measured over time and is presented as mean fold change from at least three independent experiments after normalization to the first time point within the respective group. ARP2/3-dependent actin polymerization was measured in the presence of both ERK3 and WASP (VCA) domain, and the results are depicted in *Figure 5—figure supplement 3*.

The online version of this article includes the following source data and figure supplement(s) for figure 5:

**Source data 1.** Full membrane scans for western blot images for *Figure 5B and C* and *Figure 4D*.

**Source data 2.** Prism and Excel file for *Figure 5E*.

**Source data 3.** Prism and Excel file for *Figure 5F*.

**Figure supplement 1.** ERK3 regulates F-actin levels in MDA-MB231 cells.

**Figure supplement 1—source data 1.** Full membrane scans for western blot images for *Figure 5—figure supplement 1B*.

**Figure supplement 1—source data 2.** Prism and Excel file for *Figure 5—figure supplement 1C*.

**Figure supplement 2.** ERK3 and ARP3 (ARP2/3) colocalization in CDC42-knockdown cells.

**Figure supplement 2—source data 1.** Full membrane scans for western blot images for *Figure 5—figure supplement 2A*.

**Figure supplement 2—source data 2.** Prism and Excel file for *Figure 5—figure supplement 2C and D*.

**Figure supplement 3.** ARP2/3-dependent pyrene actin polymerization in vitro was measured in the presence of full-length ERK3 (4.8 nM) or WASP (VCA) domain (400 nM), as well as in the presence of both proteins at the indicated concentrations.

**Figure supplement 3—source data 1.** Prism and Excel file for *Figure 5—figure supplement 3*.

## ERK3 phosphorylates ARP3 at S418

Conformational changes activating ARP2/3 are induced by binding to NPFs, actin, and ATP. Additionally, phosphorylation of various ARP2/3 residues has been reported previously to play a crucial role in the conformational rearrangements within the ARP2/3 complex (*Vadlamudi et al., 2004*; *Singh et al., 2003*; *Narayanan et al., 2011*; *LeClaire et al., 2008*; *Choi et al., 2013*). Most of the studies focused on phosphorylation of the ARP2 subunit, which induced the conformational repositioning of this subunit toward ARP3, further allowing binding of NPFs and full activation of the ARP2/3 complex leading to the formation of the ARP2-ARP3 heterodimer (*Narayanan et al., 2011*; *LeClaire et al., 2008*; *Choi et al., 2013*). From the seven components of the ARP2/3 complex ARP3, ARP2 and ARPC1A purified from *Acanthamoeba castellani* were shown to be phosphorylated (*LeClaire et al., 2008*). Since ERK3 is a kinase, we assessed phosphorylation of the ARP2/3 complex by ERK3 by employing in vitro kinase assays and subsequent analysis of phosphopeptides by mass spectrometry. Initial analyses detected phosphorylation of the ARP3 subunit at serine 418 (S418) (peptide sequence: HNPVFGVM**S**, please see *Supplementary file 1*). We further validated the significance of the detected phosphorylation in vivo by overexpressing the non-phosphorylatable (S418A) and phospho-mimicking (S418D) mutants of ARP3 in primary mammary epithelial cells and analyzing the cell morphology and F-actin abundance. Cells transduced with wild type (WT) ARP3 or empty vector (EV) were used as reference controls (*Figure 6F*). We were able to validate the phosphorylation of ARP3 at Ser418 in ARP3-overexpressing cells (*Figure 6G*) and detect a decrease in the S418 phosphorylation in ERK3-depleted

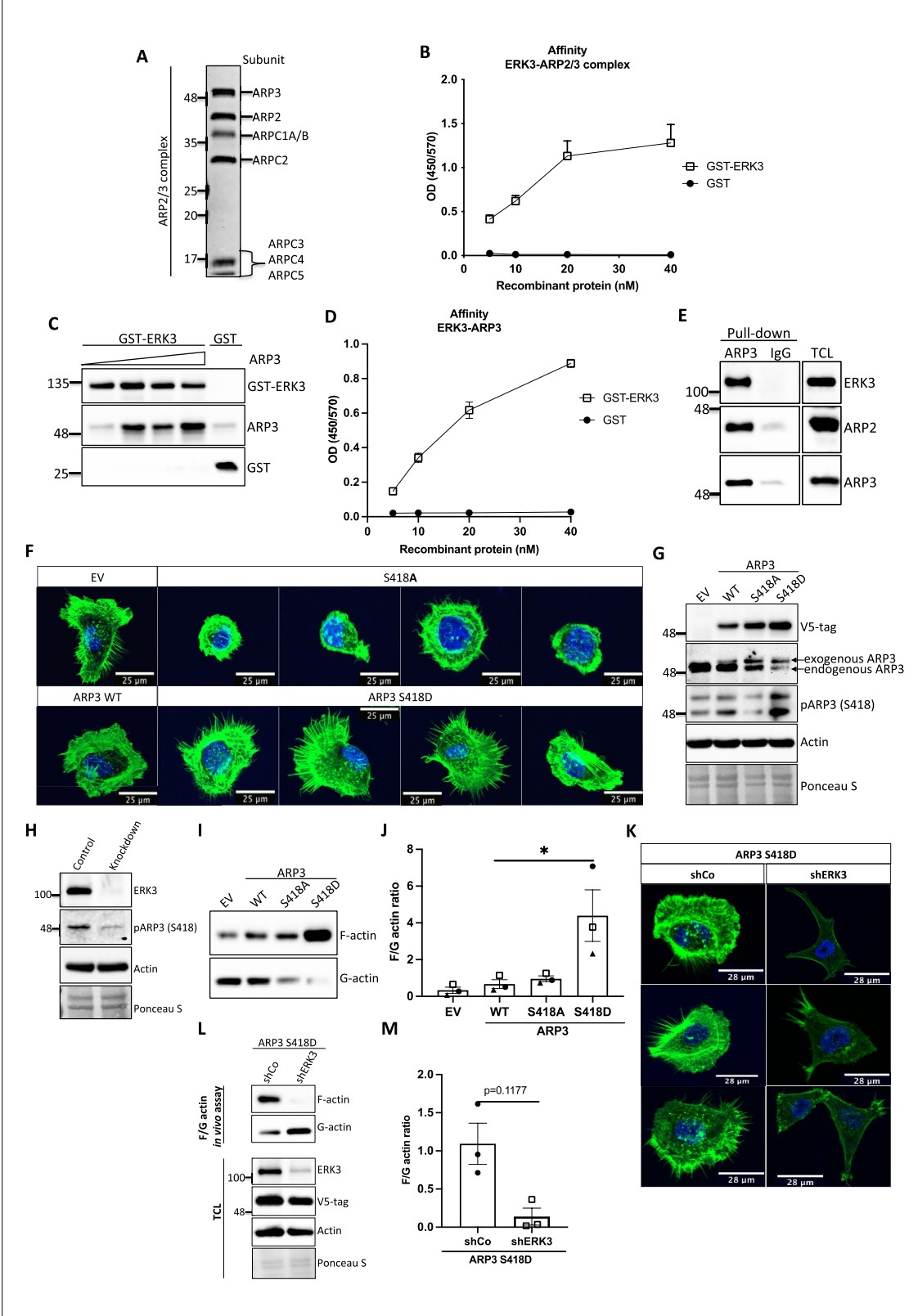

**Figure 6.** ERK3-dependent regulation of the ARP2/3 complex. (**A**) Coomassie-stained 10% SDS-PAGE gel with 1 mg of the ARP2/3 protein complex (Cytoskeleton) presenting all the subunits. (**B**) Binding of increasing concentrations of recombinant GST-ERK3 to the ARP2/3 complex was measured by ELISA as described in the 'Materials and methods' section. (**C**) The interaction between GST-ERK3 and ARP3 was measured in vitro using GST-pull-down assay as described in the 'Materials and methods' section. (**D**) Binding affinity of the recombinant GST-ERK3 protein and ARP3 was assessed

*Figure 6 continued on next page*

*Figure 6 continued*

by ELISA as described in the 'Materials and methods' section and mean absorbance (Abs) ± SEM from three independent experiments is presented. (**E**) Co-immunoprecipitation (IP) of ARP2/3 protein complex and ERK3 was performed in HMECs using ARP3 antibody. Levels of precipitated ARP3 as well as co-IP of ARP2 and ERK3 were assessed. IgG control was included to determine the specificity of the interaction. Total cell lysate (TCL) was included to present expression levels of the verified interacting partners. Ponceau S staining was used as a loading control. (**F, G**) Actin phenotype of the human mammary epithelial cells (HMECs) was validated upon stable overexpression of the ARP3 non-phosphorylatable (S418A) and the phospho-mimicking (S418D) mutant, respectively. Wild type (WT) ARP3 was used as a control for the mutants and empty vector (EV) served negative control for the overexpression itself. (**F**) F-actin expression and organization in the negative (S418A) and phospho-mimicking (S418D) ARP3 mutant was visualized by green phalloidin and merged with the Hoechst staining of the nuclei. Four representative confocal images are presented. Images of EV-transfected and ARP3 WT-overexpressing HMECs are presented as controls. (**G**) Western blot validation of the overexpression efficiency and phosphorylation of ARP3 at S418. Anti-V5-tag antibody was used to detect levels of exogenous ARP3 WT, S418A, and S418D. Expression levels of the endogenous ARP3 were assessed as well as the phosphorylation at S418, total actin was validated. Ponceau S staining was used as a loading control. (**H**) Detection of the S418 phosphorylation of ARP3 in CRISPR ERK3 HMECs presented in *Figure 5C and D*. (**I, J**) Effect of the ARP3 mutant overexpression on F-actin levels was quantified using F/G actin in vivo assay. (**I**) Representative western blot analyses of F- and G-actin levels detected in fractions obtained from EV, ARP3 WT, S418A, S418D HMECs. (**J**) Quantification of the F/G actin ratios was performed for three (n = 3) independent experiments and is presented as mean ± SEM; *p<0.0332, **p<0.0021, ***p<0.0002, ****p<0.0001, one-way ANOVA, Tukey's post-test. (**K–M**) Effect of ERK3 depletion on dense F-actin phenotype of the ARP3 S418D-overexpressing HMECs. HMECs stably overexpressing ARP3 S418D were transduced with lentiviral particles targeting ERK3 (shERK3) and stable knockdown was established as described in the 'Materials and methods' section. Cells were further subjected to analyses of the F-actin levels. (**K**) IF staining with Oregon Green Phalloidin 488 to visualize F-actin levels and organization. Scale bars 28 µm. (**L, M**) Effect of the *ERK3* knockdown on F-actin levels was quantified in the *ARP3* S418D mutant overexpressing HMECs using F/G actin in vivo assay. (**L**) Representative western blot analyses of F/G actin levels. *ARP3* S418D-(V5-tagged) overexpression and *ERK3* knockdown efficiency were validated in TCL. Actin and Ponceau S staining were used as loading controls. (**M**) Calculated ratios of F/G actin are presented as mean ± SEM from three (n = 3) independent experiments; *p<0.0332, **p<0.0021, ***p<0.0002, ****p<0.0001, paired *t*-test. Colocalization of endogenous ERK3 with endogenous and exogenous ARP3 mutant (S418D) was verified, and further effect of the *ERK3* depletion on the RAC1 and CDC42 activity was assessed in *ARP3* S418D-overexpressing HMECs and presented in *Figure 6—figure supplement 2*.

The online version of this article includes the following source data and figure supplement(s) for figure 6:

**Source data 1.** Full membrane scans for western blot images for *Figure 6A, C, E, G, I, and L*.

**Source data 2.** Prism and Excel file for *Figure 6B*.

**Source data 3.** Prism and Excel file for *Figure 6D*.

**Source data 4.** Prism and Excel file for *Figure 6J*.

**Source data 5.** Prism and Excel file for *Figure 6M*.

**Figure supplement 1.** Binding affinities of ARP3 to full-length ERK3 or ERK3 kinase domain and phosphorylation of ARP3 by wild type or kinase-dead ERK3.

**Figure supplement 1—source data 1.** Prism and Excel file for *Figure 6—figure supplement 1A*.

**Figure supplement 1—source data 2.** Full membrane scans for western blot images for *Figure 6—figure supplement 1B*.

**Figure supplement 2.** Effect of ERK3 knockdown on ARP2/3 interactions, Cdc42 and Rac1 activity, and migration of ARP3 S418D-overexpressing HMECs.

**Figure supplement 2—source data 1.** Full membrane scans for western blot images for *Figure 6—figure supplement 2A*.

**Figure supplement 2—source data 2.** Prism and Excel file for *Figure 6—figure supplement 2B*.

**Figure supplement 2—source data 3.** Prism and Excel file for *Figure 6—figure supplement 2E, F*.

**Figure supplement 2—source data 4.** Prism and Excel file for *Figure 6—figure supplement 2H*.

cells (*Figure 6H*) and in the kinase dead (KD) ERK3-overexpressing cells (*Figure 6—figure supplement 1B*). Of note, we observed that expression of exogenous ARP3 reduced the protein levels of the endogenous ARP3 (*Figure 6G*), possibly due to disruption of the protein complex, thereby affecting the stability of the endogenous proteins. Strikingly, expression of the phosphorylation-mimicking ARP3 mutant (S418D) under these settings predominantly induced actin filament formation, as indicated by the intense phalloidin staining (*Figure 6F*) and enhanced the F/G-actin ratio in primary mammary epithelial cells (*Figure 6I and J*). Moreover, S418D-overexpressing cells exhibited F-actin-rich protrusions (*Figure 6F*, lower panel). In contrast, most of the S418A-expressing cells were smaller in size and had a round morphology (*Figure 6F*, upper panel). Quantification of the F-actin levels revealed that despite the morphological distortion, cells expressing non-phosphorylatable ARP3 (ARP3 S418A) had no significant effect on overall F-actin levels (*Figure 6I and J*). To further determine the relevance of the S418 phosphorylation in ERK3-regulated ARP2/3-dependent actin polymerization in cells, we depleted ERK3 in the ARP3 S418D-overexpressing HMECs using shRNA (*Figure 6K–M*). Knockdown

of ERK3 led to a significant reduction of the F-actin levels in the ARP3 S418D-overexpressing cells (*Figure 6K–M*). These data suggested that although phosphorylation at S418 of ARP3 promoted actin polymerization and thus the F-actin content it is not absolutely essential like ERK3. A concomitant active RAC1/CDC42 pull-down revealed a significant decrease in RAC1 activity, with almost no effect on CDC42 (*Figure 6—figure supplement 2A and B*). We could still detect ARP3 and ARP2 in complex with CDC42 under these settings (*Figure 6—figure supplement 2A*). Using immunofluorescence analysis, we observed significant colocalization between endogenous ERK3 and ARP3 in HMECs (*Figure 6—figure supplement 2C*) and between endogenous ERK3 and overexpressed ARP3 S418D (*Figure 6—figure supplement 2D–F*), respectively. Further, depletion of ERK3, as expected, reduced the migration of ARP3 S418D-expressing cells (*Figure 6—figure supplement 2G and H*).

## Kinase activity of ERK3 is necessary for membrane protrusions in primary mammary epithelial cells

ERK3 controls both F-actin levels by regulation of the new filament assembly via ARP3-binding and its branching and bundling into the actin-rich protrusions by binding to CDC42/RAC1. We further investigated whether kinase activity of ERK3 was required for the formation of actin-rich protrusions and F-actin levels in mammary epithelial cells. Control (shCo) and ERK3-depleted (shERK3, 3′ UTR) HMECs were reconstituted with WT or kinase dead (KD) (K49A/K50A) ERK3, and EV was introduced as a control (*Figure 7A*). Knockdown of ERK3 led to a decrease in actin-rich protrusions and overall F-actin staining (*Figure 7B*) with concomitant decrease in F-actin abundance (*Figure 7C and D*). Complementation with ERK3 WT rescued overall F-actin levels and the protrusive phenotype of the HMECs (*Figure 7B and D*). Interestingly, KD ERK3 recovered cytoskeletal F-actin levels to the same extent as ERK3 WT (*Figure 7C and D*). However, although the abundance of F-actin is rescued upon KD ERK3 overexpression, cells formed a dense meshwork of actin membrane ruffles which did not protrude into filopodia but rather curved around the edges of the cell forming a tangled web (*Figure 7B*). These results indicate that kinase activity of ERK3 is not required for the polymerization of actin, but rather for the bundling and/or branching of the rapidly polymerizing actin filaments. To further corroborate these observations, the WT and KD ERK3 proteins were expressed in the rabbit reticulocyte lysate (RRL) system and then employed to stimulate ARP2/3-dependent actin polymerization in vitro. Kinase activity of ERK3 did not seem to affect the ARP2/3-dependent actin polymerization in vitro (*Figure 7—figure supplement 1*). We performed active RAC1/CDC42 pull-down from HMECs shERK3 (3′ UTR) cells reconstituted with either WT or the KD ERK3 and found that the kinase activity of ERK3 was not required for interactions or activation of CDC42. However, RAC1 activation was partially dependent on the kinase activity of ERK3 (*Figure 7E–I*).

## Discussion

The regulation of the actin cytoskeleton is an intensely investigated area because of its fundamental role in many basic cellular functions. The signaling machinery that controls the actin skeleton dynamics is deregulated in cancer contributing to metastasis. Kinases are major drivers of signaling events and form the largest part of the druggable genome. Several drugs have successfully been developed to target deregulated kinases in cancer. Of the more than 500 kinases encoded by the human genome, many studies have been focused on the conventional MAPKs, while the pathophysiological significance of atypical MAPKs remains underexplored. We have recently demonstrated that ERK3 directly contributed to AP-1/c-Jun activation, IL-8 production, and chemotaxis (*Bogucka et al., 2020*). Emerging studies have further shown that ERK3 functions in a context- and tissue-specific manner in controlling tumorigenesis and metastasis (*Al-Mahdi et al., 2015*; *Bogucka et al., 2021*; *Elkhadragy et al., 2020*; *Elkhadragy et al., 2017*; *Choi et al., 2013*; *Elkhadragy et al., 2018*; *Alshammari et al., 2021*). Mice expressing a catalytically inactive ERK3 mutant survive to adulthood and are fertile, but the kinase activity of ERK3 is necessary for optimal postnatal growth. In this study, we aimed to elucidate how ERK3 influences cell shape and actin cytoskeleton and unexpectedly revealed a pivotal role for this atypical MAPK in the regulation of RhoGTPases as GEF, of ARP2/3-dependent polymerization of actin probably as NPF, and of cell migration in mammalian cells (*Figure 8*). Further, our studies unveiled an evolutionarily conserved role for ERK3 in the formation of actin-rich protrusions.

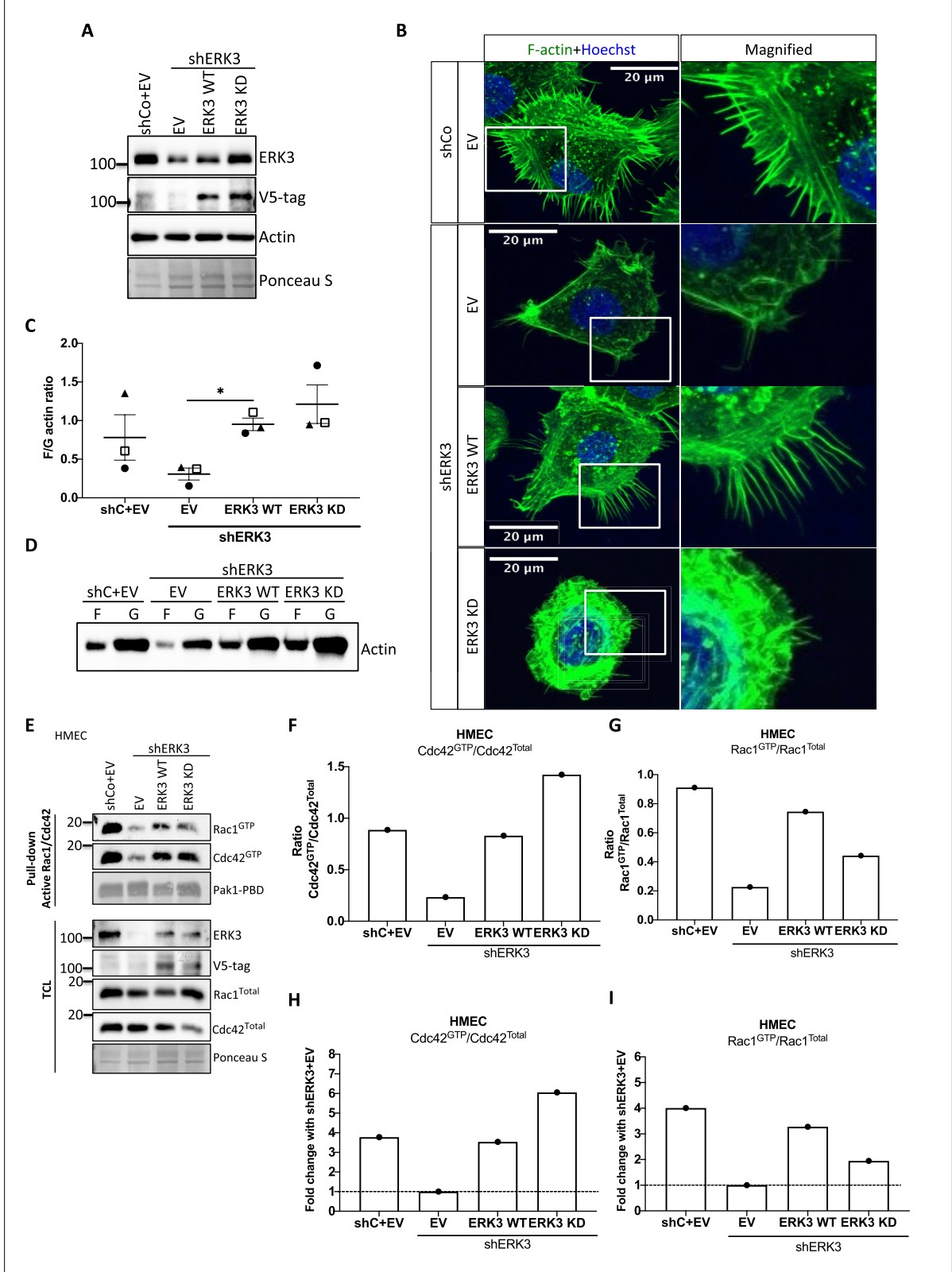

**Figure 7.** Kinase dead (KD) mutant of ERK3 inhibits actin-rich protrusions with no negative effect on F-actin levels. (**A**) Western blot validation of the ERK3 levels and expression of the V5-tagged *ERK3* constructs is presented as well as the total actin levels. Ponceau S was used as a loading control. (**B**) Confocal analyses of control (shCo EV) and *ERK3*-depleted (sh*ERK3*) human mammary epithelial cells (HMECs) reconstituted with wild type (ERK3 WT) or KD (ERK3 KD) (K49A/K50A) mutant of *ERK3* after 4 hr starvation. Merged images of F-actin (green, phalloidin) and Hoechst are shown on the left

*Figure 7 continued on next page*

*Figure 7 continued*

representing cell phenotype and magnified regions of the cell edges on the right show the actin distribution. (**C, D**) Levels of F-actin in HMECs analyzed by phalloidin staining were quantified using a F/G actin in vivo assay. (**C**) Calculated ratios of F/G actin are depicted as mean ± SEM from three (n = 3) independent experiments; *p<0.0332, **p<0.0021, ***p<0.0002, ****p<0.0001, one-way ANOVA, Tukey's post-test. (**D**) Representative western blot analyses of F- and G-actin levels. (**E–I**) Active RAC1 and CDC42 pull-down assays were performed from control (shCo) and *ERK3*-depleted (sh*ERK3*, 3' UTR) HMEC reconstituted with either WT or KD *ERK3* as described in the 'Materials and methods' section. PAK1-PBD was used to capture the active form of CDC42 and RAC1 in the respective cell lysates. Levels of active (GTP-bound) CDC42 and RAC1, as well as the total protein expression were assessed, *ERK3* knockdown efficiency and overexpression was verified in the total cell lysate (TCL) using ERK3 antibody and V5-tag antibody to detect the exogenous WT and KD version of ERK3. (**E**) Western blot analyses are presented. (**F, G**) Relative levels of active CDC42 and RAC1 were calculated with respect to the total protein levels and are presented as ratio of GTP/Total RhoGTPases. (**H, I**) Additionally, the ratio of GTP/total RAC1 and CDC42 was normalized to the control cells (sh*ERK3*+EV) and is presented as fold change in activation.

The online version of this article includes the following source data and figure supplement(s) for figure 7:

**Source data 1.** Full membrane scans for western blot images for *Figure 7A, D, and E*.

**Source data 2.** Prism and Excel file for *Figure 7C*.

**Source data 3.** Prism and Excel file for *Figure 7F–I*.

**Figure supplement 1.** Effect of RRL-expressed wildtype or kinase-dead ERK3 on ARP2/3-dependent pyrene actin polymerization.

**Figure supplement 1—source data 1.** Full membrane scans for western blot images for *Figure 7—figure supplement 1A*.

**Figure supplement 1—source data 2.** Prism and Excel file for *Figure 7—figure supplement 1B*.

## ERK3 is a GEF for CDC42

The activation of both RAC1 and CDC42 was compromised in the absence of ERK3 in primary mammary epithelial cells and oncogenic MDA-MB-231 cells. While the activation of RAC1 was partially rescued in HMECs upon EGF stimulation in ERK3-depleted cells, this was not the case in MDA-MB231 cells (*Figure 2B, C, F, and G*). Furthermore, while ERK3 regulated the activity of both RAC1 and CDC42 in vivo (*Figure 2B–G*), the kinase domain of ERK3, which lacks kinase activity, directly facilitated GDP-GTP nucleotide exchange specifically for CDC42 (*Figure 3B, C, and G*), which is implicated as a main regulator of filopodia assembly. In some cell types, we often detect a reduction in the total protein levels of RAC1 and CDC42 after depletion of ERK3. Whether inactivation of these RhoGTPases prompts them for proteasomal degradation under these settings deserves further analysis. Using several experimental procedures, we demonstrated that the interaction between ERK3 and RAC1 or CDC42 is direct and nucleotide-independent, which suggests that ERK3 is probably targeting the C-terminus of these RhoGTPases. Interestingly, it has been reported that RAC1 possesses a C-terminal docking (D site) for the canonical ERK ([183]KKRKRKCLLL[192]) (*Tong et al., 2013*). Whether the same site is being exploited by ERK3 for its binding to RAC1 or CDC42 deserves further studies. It is interesting to note that while the kinase domain of ERK3 promoted GTP binding to CDC42, it slightly attenuated the same process with RAC1 (*Figure 3B–G*). While the kinase activity of ERK3 is not required for CDC42 activation, RAC1 activity partially depends on it. Further structural, biochemical, and biophysical studies are clearly warranted to unveil the molecular basis of this interaction and the subsequent functional implications.

In cells, regulation of RhoGTPases and that of the actin assembly is a much more complex and integrative process (*Kurokawa et al., 2004*). For example, a cell type-dependent, hierarchical crosstalk between RhoGTPases was observed in the regulation of actin-rich protrusions and CDC42 activity was shown to initiate RAC1-dependent lamellipodia protrusions (*Nobes and Hall, 1995*; *Zamudio-Meza et al., 2009*; *Nobes and Hall, 1999*). While ERK3 might activate RAC1 through its GEF-like function toward CDC42, it could also function as an intermediate link between RAC1 and its GEF by scaffolding both proteins and/or activating the GEF, thus indirectly contributing to RAC1 activity.

It is highly interesting that PAK family kinases, the effector kinases of RAC1 and CDC42, directly phosphorylate the SEG motif in the activation loop of ERK3 (*Déléris et al., 2011*; *De la Mota-Peynado et al., 2011*). The observations presented here strongly suggest a positive feedback loop, where ERK3 is required for the GTP loading of CDC42 and RAC1 to induce their interaction with PAKs, which in turn phosphorylate and activate ERK3. We show that active ERK3 kinase is required for the eventual formation of actin-regulated membrane protrusions. The functional and physical uncoupling of this loop could possibly be controlled by a phosphatase or by inducing the rapid turnover of ERK3 protein. The constituents of this dynamic complex need further evaluation and characterization.

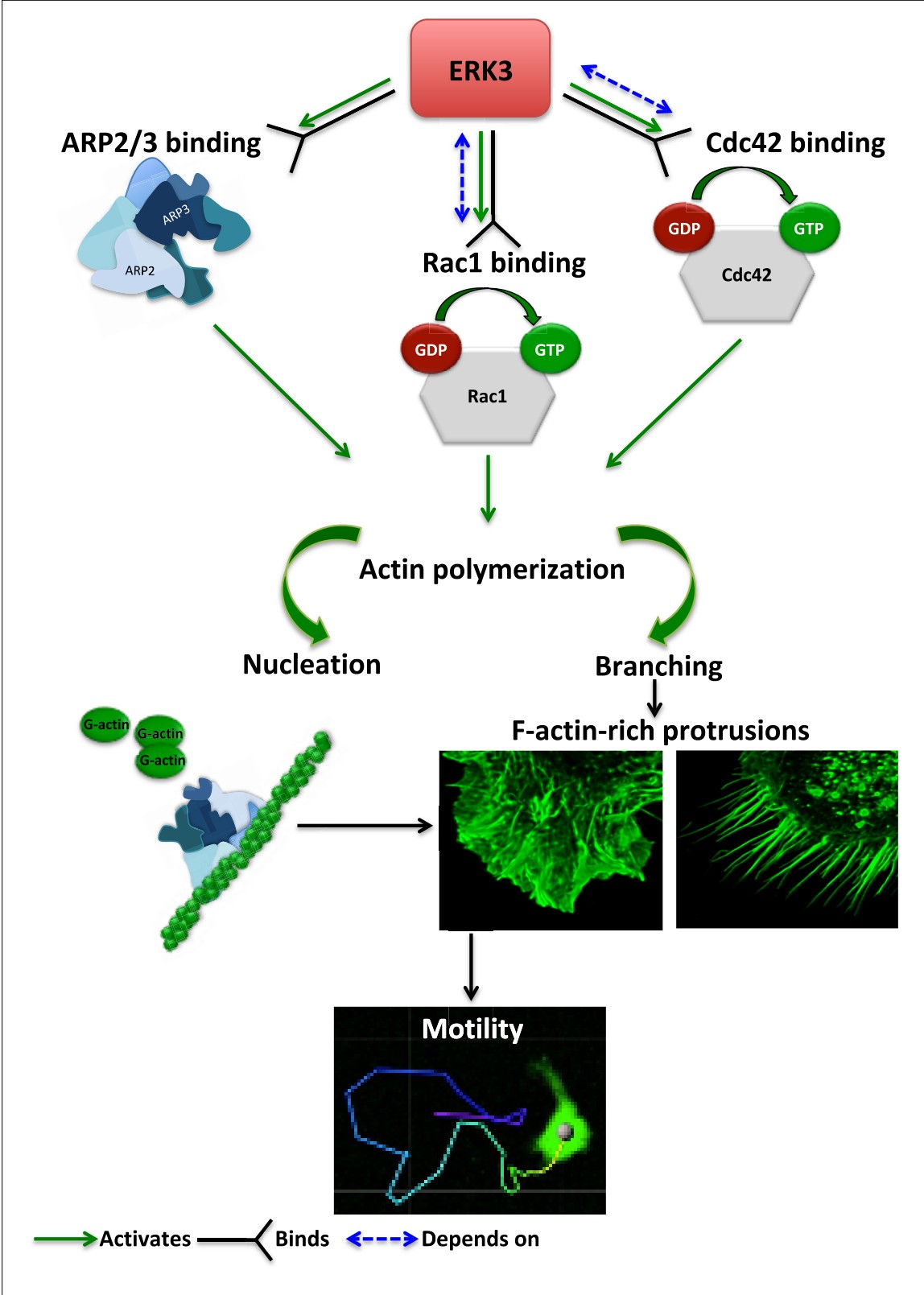

**Figure 8.** Schematic summary of the ERK3-dependent mechanisms regulating actin cytoskeleton and cell motility. ERK3 directly binds and activates the ARP2/3 protein complex as well as the CDC42 and RAC1 Rho GTPases. Activation of the ARP2/3 complex and RAC1/CDC42 is required for nucleation of the new actin filaments, elongation, and branching into the lamellipodia and filopodia. ERK3 regulates actin-rich protrusions, which play a direct role in cell motility.

# ERK3 binds ARP3 to regulate its function in actin polymerization

Cells rely on several mechanisms to regulate the assembly of actin filaments. To complicate matters further, regulatory pathways involved in the filopodia formation seem to be cell type-specific, and CDC42-, WASP-, and ARP2/3-independent mechanisms of membrane spikes formation have been proposed (*Czuchra et al., 2005*; *Snapper et al., 2001*; *Steffen et al., 2006*). In this study, we demonstrate that ERK3 directly binds the ARP2/3 complex and contributes to actin polymerization (*Figures 5 and 6*). Conformational changes in ARP2/3 complex assembly are crucial for the initial heterodimer formation between ARP2 and ARP3 subunits (*Machesky et al., 1999*; *Gournier et al., 2001*; *Welch et al., 1997*). In the unstimulated complex, ARP2 and ARP3 subunits exist in a splayed confirmation and upon stimulation by NPFs the two proteins align into a side-by-side position creating a surface mimicking the first two subunits of the new actin filament (*Robinson et al., 2001*; *Mullins and Pollard, 1999*). Binding of these two crucial subunits by NPFs could reassure an active conformation. Interestingly, phosphorylation of certain subunits of ARP2/3 complex has been shown to be indispensable for the destabilization of its inactive conformation prior the full activation by the NPFs (*Vadlamudi et al., 2004*; *Narayanan et al., 2011*; *LeClaire et al., 2008*; *Choi et al., 2013*). LeClaire et al. further reported that Nck-interacting kinase (NIK) regulates ARP2 phosphorylation and grooms the ARP2/3 complex for the activation by NPFs. Phosphorylation events within the ARP2/3 complex could regulate nucleation activity by affecting several properties, such as complex formation itself, conformational changes in the complex assembly, or the affinity of the ARP2/3 to NPFs. Further structural simulation studies are needed to test whether the interaction of ERK3 with the ARP2/3 complex and ARP3 phosphorylation at S418 inflicts key conformational changes to support actin polymerization. While we uncover a role for ERK3 in the formation of filopodia and ARP2/3-dependent actin polymerization, the possible role for ERK3 in the regulation of other actin nucleation factors including formins cannot be ruled out.

While the kinase activity of ERK3 is clearly required for the formation of actin-rich protrusions, expression of KD constructs in ERK3-depleted cells failed to inhibit F-actin enrichment. These data suggest that perhaps ERK3 binding to ARP2/3 and possibly other hitherto unknown factors per se contribute to actin polymerization, while its kinase activity is probably still required for ARP2/3-dependent and CDC42-induced filopodia formation, polarization, and migration of mammalian epithelial cells.

Overall, our studies shed new light on the multilayered regulation of the actin cytoskeleton by an understudied MAPK (*Figure 8*). ERK3 is druggable, and kinase inhibitors have already been developed for clinical use, which could also serve as tools to decipher kinase-dependent and -independent functions. Our observations will not only enhance the current understanding of actin cytoskeleton regulation, but also instigate new lines of investigations on the molecular, structural, and functional characterization of RhoGTPases and the ARP2/3 complex.

## Materials and methods

### Cell culture

Immortalized HMECs hTERT-HME1 (ME16C) (ATCC CRL-4010) were purchased from ATCC (Manassas, VA) and cultured in mammary epithelial cell growth medium (MEGM) BulletKit, containing supplements and growth factors (Cat# CC-3150, Lonza) according to the Clonetics mammary epithelial cell system guide (Lonza). Triple-negative breast cancer cell line MDA-MB231 was purchased from DSMZ (DSMZ# ACC 732) and cultured in Dulbecco's Modified Eagle Medium (DMEM) supplemented with 10% heat-inactivated FBS. MDA-MB231 cells overexpressing GFP actin were a kind gift from Prof. Dr. Mohamed Bentires-Alj (University of Basel, Department of Biomedicine). 293T cells were a kind gift from Dr. Andreas Ernst (Goethe-University Frankfurt am Main, IBC2) and were cultured in DMEM supplemented with 10% FBS. MDA-MB231 cells overexpressing GFP were a kind gift from Prof. Dr. Mohamed Bentires-Alj (University of Basel, Department of Biomedicine). MDA-MB468 cells were purchased from DSMZ and cultured in DMEM/F-12 medium (Cat# 11320033, Gibco) supplemented with 10% FBS. MCF7 cells were cultured in Roswell Park Memorial Institute 1640 (RPMI 1640) medium (Cat# 11875093, Gibco) supplemented with 10% FBS, 1% MEM non-essential amino acids (Cat# M7145, Sigma), 1 mM sodium pyruvate, and 10 μg/ml human insulin (Cat# 19278, Sigma). HUVECs were a kind gift from Prof. Dr. Wolfram Ruf (Center for Thrombosis and Hemostasis [CTH], University

Medical Center of the Johannes Gutenberg University Mainz) and were cultured on 0.2% gelatin coating in Endothelial Cell Growth Medium (Cat# C-22010, PromoCell). HT-29 cells were purchased from ATCC (ATCC HTB-38) and cultured in McCoy's medium supplemented with 10% heat-inactivated FBS. Calu-1 cells were obtained from Sigma and cultured in DMEM supplemented with 10% heat-inactivated FBS.

All cell lines used in this study were authenticated cell lines obtained from ATCC or DSMZ. All used cells were periodically tested for *Mycoplasma* contamination with negative results.

## Stimulation of cells

HMECs and MDA-MB231 cells were seeded in 12-well plates at an initial density of $2 \times 10^5$ cells/well. After cells reached 70% confluence, medium was exchanged to MEGM with no supplements for HMECs and DMEM minus FBS for MDA-MB231 cells, 4 hr prior treatment with recombinant human epidermal growth factor (EGF) (Cat# RP-10927, Invitrogen) at 5, 10, 15, and 30 min. Afterward, cells were subjected to western blot analyses.

## Cycloheximide (CHX) chase experiments

To determine ERK3 half-life in mammary epithelial cells and its alteration upon EGF treatment, HMECs were seeded in 12-well plates at an initial density of $2 \times 10^5$ cells/well and cultured until 70% confluent. Medium was exchanged to MEGM with no supplements, 4 hr before cells were treated with human recombinant EGF at 100 µg/ml for 15 min, followed by treatment with protein synthesis inhibitor CHX (Cat# C-7698, Sigma, stock 100 mg/ml in DMSO) at 0.5, 1, 2, 4, and 6 hr. Cells were subjected to western blot analyses and quantification. Fold change in ERK3 protein levels was calculated with respect to the untreated cells (-EGF, 0 hr) or to the respective control in each group (0 hr) for unstimulated (-EGF) and EGF-stimulated (+EGF) cells using ImageJ software.

## Antibodies

Primary antibodies used: phospho-ERK3 (pSer189) (Cat# SAB4504175, Sigma), ARPC1A (Cat# HPA004334, Sigma), ERK3 (Cat #4067, Cell Signaling Technology [CST]), phospho-p44/42 MAPK (Thr202/Tyr204) (Cat# 9101L, CST), V5-tag antibody (Cat # R960-25, Invitrogen), GST antibody (Cat# 2622S, CST), normal rabbit IgG antibody (Cat# 2729, CST), GST (B-14) antibody (Cat# sc-138, Santa Cruz Biotechnology), RAC1 (Cat# 610651, BD Transduction Laboratories), CDC42 (Cat# 610929, BD Transduction Laboratories), and ARP3 (phospho-Ser418) antibody (Cat# orb317559) were obtained from Biorbyt and their specificity validated in cells overexpressing active (S418D) and phospho-impaired (S418A) ARP3 mutants (*Figure 6G*). ARP3 (Cat# ab151729, Abcam), ARP2 (Cat# ab128934, Abcam), and beta-actin HRP-conjugated antibody (Cat# ab49900, Abcam). HRP-conjugated secondary antibodies for rabbit and mouse IgG were obtained from Invitrogen (Cat# A16096 and A16066, respectively).

Antibodies used for the IF staining: ERK3 antibody (Cat# MAB3196, R&D).

## Western blotting

Cells were washed with ice-cold PBS (10 mM sodium phosphate, 150 mM NaCl, pH 7.2) and lysed in cold RIPA lysis buffer: 250 mM NaCl, 50 mM Tris (pH 7.5), 10% glycerol, 1% Triton X-100, supplemented with protease inhibitor cocktail Set I-Calbiochem 1:100 (Cat# 539131, Merck Millipore) and phosphatase inhibitors: 1 mM sodium orthovanadate ($Na_3VO_4$), 1 mM sodium fluoride (NaF). Cells were lysed for 30 min on ice, followed by 10 min centrifugation at 14,000 rpm. Protein concentrations were estimated using 660 nm Protein Assay (Cat# 22660, Thermo Fisher Scientific). Samples were prepared by mixing with 4× SDS-PAGE sample buffer (277.8 mM Tris-HCl pH 6.8; 44.4% glycerol, 4.4% SDS, 0.02% bromophenol blue) supplemented with 50 mM dithiothreitol (DTT) and boiling at 95°C for 5 min. Samples were then subjected to SDS-PAGE followed by transfer of the proteins onto nitrocellulose membranes (GE Healthcare, Chalfont St Giles, UK). Membranes were blocked in 3% BSA/PBST (1× PBS, pH 7.5 containing 0.5% Tween-20) for 1 hr at room temperature (RT) and incubated with primary antibody diluted in PBST at 4°C, overnight. Following 3 × 5 min washing with PBST, membranes were incubated with HRP-conjugated secondary antibody for 1 hr at RT. After washing, signal was visualized using chemiluminescent HRP substrate (Immobilon Western, Cat# WBKLS0500,

Merck Millipore). Western blots semi-quantification was performed using ImageJ software (ImageJ, RRID:SCR_003070).

## Site-directed mutagenesis and expression constructs

Full-length *ERK3* WT pDONR-223 construct purified from Human Kinase Library (Addgene) was used as a template to generate *ERK3* K49A K50A KD mutant. Mutations were introduced using the following primers were used: frw_5′ GCAATTGTCCTTACTGATCCCCAGAGTGTC, rev_5′ CGCGATGG CTACTCTTTTGTCACAGTC using Q5 High-Fidelity DNA Polymerase (Cat# MO491, New England BioLabs). For lentiviral expression, *ERK3* WT and *ERK3* K49A K50A (KD) mutant were cloned into pLenti4TO/V5-Dest expression vector.

WT *ARP3* pENTR223 construct (Harvard HsCD00375598) was cloned into pLenti4TO/V5-Dest mammalian expression vector and pGEX-6-P1 for bacterial expression. Serine 418 (S418) phopsho-dead (S418A) and phosphor-mimicking (S418D) constructs of *ARP3* were obtained from Eurofins (Cat# #11104144057-1-5 and #11104144057-1-6, respectively) and cloned into the pLenti4TO/V5-Dest expression vector.

## Generation of knockdowns

shRNAs plasmids were purified from the MISSION shRNA Human Library (Sigma). Non-targeting control shRNA (shCo) MISSION pLKO.1 puro (Cat# SHC001) was included.

> sh*MAPK6* (sh*ERK3*) TRCN0000001568, NM_002748.x-3734s1c1, CCGGGCTGTCCACGTA CTTAATTTACTCGAGTAAATTAAGTACGTGGACAGCTTTTT
> sh*CDC42*#1 TRCN0000047628, NM_001791.2–471s1c1 CCGGCCCTCTACTATTGAGAAACT TCTCGAGAAGTTTCTCAATAGTAGAGGGTTTTTG
> sh*CDC42*#2 TRCN0000047629, NM_001791.2–193s1c1 CCGGCGGAATATGTACCGACTGTT TCTCGAGAAACAGTCGGTACATATTCCGTTTTTG

siRNAs were purchased from QIAGEN:
si*MAPK6*#1 (si*ERK3*#1) FlexiTube siRNA 5 nmol, siRNA Name: Hs_*MAPK6*_5, Cat# SI00606025, sense strand: 5′- AGUUCAAUUUGAAAGGAAATT-3′. Negative control siRNA (siCo) Cat# 1027310.

## siRNA transfection

Cells were seeded 1 d before transfection at an initial density of $2 \times 10^5$ cells/well in 12-well plates or at $3 \times 10^5$ cells/well (6-well plates). Cells were transfected using SAINT-sRNA transfection reagent (SR-2003, Synvolux) according to the manufacturer's instructions. Cells were analyzed 48 hr post-transfection, and knockdown efficiency was verified by western blot using target-specific antibodies.

If EGF stimulation was included in the experimental settings, 48 hr post-transfection medium was exchanged for serum/supplements-free medium for 4 hr prior the EGF (100 ng/ml) treatment for 15 min.

## Lentiviral-mediated knockdown and expression of cDNAs

For the production of lentiviral particles, the following packaging plasmids were used: pHDM-G (encoding VSV-G), pHDM Hgpm2 (encoding codon-optimized HIV gag-pol proteins), pHDM tat 1b (encoding HIV Tat1b protein), and pRC CMV-Rev1b (encoding HIV rev protein). Lentiviral particles were generated following standard protocols. Briefly, lentiviral supernatants were produced in 293T cells by co-transfection of the cells with lentiviral packaging plasmids (0.3 µg each), lentiviral expression constructs (1 µg), and 10.8 µl of 10 mM polyethylenimine (PEI). The viral particles were harvested after 48 hr and sterile-filtered. Cells were infected with lentiviral particles in the presence of 10 µg/ml of polybrene (Cat# sc-134220, Santa Cruz) and selected with puromycin at 3 µg/ml (Cat# 0240.3, Carl Roth).

For complementation assays, EV control (pLenti4TO/V5-Dest), WT, or KD (K49A K50A) mutant of *ERK3 (MAPK6)* were reintroduced into sh*ERK3* (3′UTR) background by lentiviral transduction and cells were double-selected with zeocin (100 µg/ml) (Cat# R25001, Invitrogen) and puromycin.

To generate CRISPR/Cas9-mediated *ERK3 (MAPK6)* knockout in the HMECs, cells were infected with lentiviral particles and selected with puromycin (8 µg/ml). Lentiviral particles coding for *CRISPR ERK3* and *CRISPR* control vector (pLentiCRISPRv2) were produced in 293T cells by co-transfection of

lentiviral packaging plasmids (0.3 µg each) and 1.1 µg of lentiviral vector containing the respective gRNAs in the presence of 21 µl of Lipofectamine2000 (Cat# 11668027, Thermo Fisher Scientific).

CRISPR/Cas gRNA sequences targeting *ERK3* were designed by Rule Set 2 of Azimuth 2.0 as described previously (*Doench et al., 2016*). The top three scoring gRNAs were selected:

> #1 5′-CACCGAGCCAATTAACAGACGATGT-3′
> #2 5′-CACCGATACTTGTAACTACAAAACG-3′
> #3 5′-CACCGCTGCTGTTAACCGATCCATG-3′

gRNAs were individually cloned into pLentiCRISPRv2 (Addgene plasmid #52961), following established protocols (*Sanjana et al., 2014*).

## Transient transfections

HMEC cells stably transfected with shRNA targeting *ERK3* at 3′UTR (sh*ERK3*) or with control EV shRNA (shCo) were transiently transfected with either an EV pcDNA3/V5-Dest40 (EV), *ERK3* WT, or *ERK3* K49A K50A, *ERK3* S189A or S189D mutant construct (0.5 µg plasmid) in the presence of PEI (3 µl/well). 6 hr post-transfection medium was exchanged for DMEM + FBS complete medium. 24 hr post-transfection medium was exchanged again for DMEM–FBS medium. 48 hr post-transfection supernatants were harvested for IL-8 ELISA and cells were analyzed by western blot.

The images of both GFP fluorescence and transmission of the live cells were acquired with a Leica SP8 confocal microscope (Leica, Mannheim, Germany) performed with a 10 × 0.3 NA objective, with 488 nm excitation (at approximately 150 µW), with an emission window of 500–590 nm for GFP detection and with scanning Differential Interference Contrast transmission imaging in a 1500 µm × 1500 µm frame format with 400 lines per second, 0.71 µm/pixel (2048 × 2048 pixel per frame) and with two times averaging per line with a frame acquisition of every 30 min per selected position within the chamber.

Phosphorylation of the ARP2/3 protein complex subunits by ERK3 was detected by mass spectrometry analyses after in vitro kinase assay. Proteins were in-gel-digested and subsequently enriched for phosphopeptides using $TiO_2$ beads.

## RNA isolation, cDNA synthesis, and RT-PCR analysis

For gene expression analyses, cells were washed with cold PBS and total RNA was extracted using Trizol (Cat# 15596018, Ambion) according to the manufacturer's instructions. Isolated RNA was then used as a template for cDNA synthesis with the RevertAid First Strand cDNA synthesis kit (Cat# K1621, Thermo Fisher Scientific) and random hexamer primers.

Real-time PCR was performed using EvaGreen qPCR master mix (5x Hot Start Taq EvaGreen qPCR Mix [No ROX], Cat# 27490, Axon) and the following primers:

> *ERK3* Frw_5′ ATGGATGAGCCAATTTCAAG
> Rv_5′ CTGACAATCATGATACCTTTCC

The housekeeping gene for human 18S was used for normalization:

> *18s* Frw_5′ AGAAACGGCTACCACATCCA
> Rv_5′ CACCAGACTTGCCCTCCA;

Relative expression levels were calculated as ΔΔCt, and results are presented as log2fold change in gene expression.

## Protein purification

The pGEX-2T-*RAC1*-WT and pGEX-2T-*CDC42*-WT bacterial expression constructs were a gift from Gary Bokoch (Addgene plasmid# 12977 and #12969, respectively). The pGEX2TK-*PAK1* (70–117) containing the p21-binding domain (PBD) was a gift from Jonathan Chernoff (Addgene plasmid# 12217). For the purification of ARP3 (ACTR3), construct (Harvard HsCD00375598) in pGEX-6-P1 expression vector was used.

Rosetta (DE)-competent cells expressing *RAC1*-WT, *CDC42*-WT, or *ARP3* were incubated at 37°C, followed by induction of protein synthesis with 0.1 mM isopropyl β-D-1-thiogalactopyranoside (IPTG) at 16°C, overnight. Bacteria were lysed in 1% NP-40, 20 mM Tris–HCl, pH 7.5, 200 mM NaCl buffer.

Sonicated lysates were used for the GST purification using PureCube glutathione agarose (Cat# 32105, Cube Biotech). After incubation, beads were washed with PBS pH 7.5.

Enzymatic cleavage of the GST tag was performed using thrombin (Cat# T1063, Sigma). Supernatants containing cleaved target proteins were incubated with Benzamidine Sepharose 4 flat flow (Cat# 17-5123-10, Cytvia) to remove thrombin. If required, eluted proteins were then concentrated using amicon ultra-15 centrifugal filters (Cat# UFC901024, Merck Millipore).

Proteins size and purity were assessed throughout the purification procedures.

## Synthesis of WT and KD ERK3 proteins using RRL

*ERK3* WT or *ERK3* KD (K49A K50A) in pcDNA3/V5-Dest40 vector were expressed in T7 RRLs using TNT T7 Quick Coupled transcription/Translation System (Cat# TM045, Promega) following the manufacturer's protocol. Expression levels were assessed by western blot, using V5-tag specific antibody. Lysates were further used in actin polymerization assay as recombinant proteins.

## Endogenous pull-down of active RAC1/CDC42

Levels of GTP-bound RAC1 and CDC42 were determined using either active RAC1/CDC42 pull-down and detection kit (Cat# 16118/19, Thermo Fisher Scientific) according to he manufacturer's instructions and buffer composition or purified GST-PAK1-PBD fusion beads. HMEC and MDA-MB231 cells were seeded in 6-well plates at an initial density of $3 \times 10^5$ cells/well. After cells reached 70% confluence, medium was exchanged to MEGM (no supplements) for HMEC or DMEM without FBS for MDA-MB231 cells, 4 hr prior 15 min stimulation with 100 ng/ml of recombinant human EGF.

Afterward, cells were subjected to active RAC1/CDC42 pull-down. The precipitated samples and total cell lysates were subjected to western blot analyses and ImageJ quantification. Relative levels of active RAC1 and CDC42 were determined by calculating the ratio of active (GTP loaded) RhoGTPases with respective total protein levels in TCL.

## Endogenous pull-down of ARP3

HMECs were seeded in 6-well plate at an initial density of $2 \times 10^5$ cells per well and cultured until 80% confluent. For the immunoprecipitation (IP) of the endogenous ARP3, cells were washed with ice-cold PBS and lysed with ice-cold IP buffer (10 mM HEPES pH 7.4; 150 mM NaCl, 1% Triton X-100, plus protease inhibitor cocktail Set I-Calbiochem 1:100 [Cat# 539131, Merck Millipore], 1 mM $Na_3VO_4$, and 1mM NaF). Cell lysates were incubated with Protein A/G-Agarose beads (Cat# 11 134 515 001/11 243 233 001, Roche), and either ARP3 or normal rabbit IgG antibody for 2 hr at 4°C with rotating. After the incubation, beads were washed with IP buffer and analyzed by immunoblot. Levels of the immunoprecipitated ARP3 as well as the co-immunoprecipitation of ARP2 and ERK3 were assessed. Total cell lysates were used as a control.

## Binding interaction-GST pull-down assays

Purified recombinant GST-fusion RAC1 and CDC42 proteins immobilized on the beads were used for in vitro GST pull-down experiments.

To verify the relevance of the nucleotide binding in the interaction of the GTPases with ERK3 protein, RAC1 and CDC42 proteins were loaded with non-hydrolyzable GTPγS or GDP (components of the active RAC1/CDC42 pull-down and detection kit; Cat# 16118/19, Thermo Fisher Scientific).

Recombinant GST-fusion GTPases protein beads were incubated with gentle shaking for 15 min at 30°C in 100 μl of 25 mM Tris-HCl, pH 7.2, 150 mM NaCl, 5 mM $MgCl_2$, 1% NP-40, and 5% glycerol binding/wash buffer containing 0.1 mM of GTPγS or 1 mM GDP in the presence of 10 mM EDTA pH 8.0 to facilitate nucleotide exchange. Reaction was terminated by placing samples on ice and addition of 60 mM of $MgCl_2$. GTPγS-/GDP-loaded RAC1 and CDC42 beads were centrifuged and supernatants were removed. Beads were subjected to GST pull-down ERK3 binding assay.

GTPγS/GDP-loaded GST-RAC1 and GST-CDC42 protein beads were incubated with recombinant human ERK3 protein (Cat# OPCA01714, Aviva Systems Biology) in 100 μl of binding/wash buffer supplemented with protease inhibitor with protease inhibitor cocktail Set I-Calbiochem 1:100 (Cat# 539131, Merck Millipore) and phosphatase inhibitors: 1 mM sodium orthovanadate ($Na_3VO_4$), 1 mM sodium fluoride (NaF) for 1 hr at 4°C with rotation. Recombinant GST protein bound to glutathione beads was used as a negative control. After the incubation, beads were washed three times with 400

µl binding/wash buffer, centrifuged each time at 2000 rpm. Samples were eluted with 4× Laemmli buffer supplemented with 100 mM DTT and 5 min boiling at 95°. Samples were subjected to western blot analyses.

For the in vitro interaction between GST-RAC1/ GST-CDC42 and ERK3 kinase domain, RAC1/ CDC42 WT beads and ERK3 (amino acids (aa) 9–327) (Crelux) *Escherichia coli* purification was employed. Additional information on the Crelux protein can be viewed in the *Supplementary file 1*.

For ERK3-ARP3 binding studies, human recombinant GST-fusion ERK3 protein (SignalChem) and *E. coli* purified ARP3. Glutathione beads were incubated with 37 nM of GST-ERK3 or GST alone and indicated concentrations of ARP3 for 2 hr at 4°C in the binding buffer (5 mM Tris-HCl, pH 8.0, 0.2 mM CaCl$_2$) supplemented with protease and phosphatases inhibitors. Afterward, beads were washed three times with 400 µl binding buffer supplemented with 1% NP-40. Samples were eluted with 4× Laemmli buffer supplemented with 50 mM DTT and 5 min boiling at 95°. Immobilized protein complexes were detected by GST and ARP3-specific antibodies.

## Determination of GTP loading on Rho GTPases

### In vitro active RAC1/CDC42 pull-down assay

The efficiency of the nucleotide loading under chosen conditions of the GTP-loading status of the purified RAC1 and CDC42 proteins was determined using GST-PAK1-PBD fusion beads.

Recombinant RAC1 and CDC42 proteins (no tag) were loaded with GTPγS or GDP by incubation for 15 min at 30°C in 100 µl of 25 mM Tris-HCl, pH 7.2, 150 mM NaCl, 5 mM MgCl$_2$, 1% NP-40, and 5% glycerol binding/wash buffer containing 0.1 mM of GTPγS or 1 mM GDP in the presence of 10 mM EDTA pH 8.0. Reaction was terminated by placing samples on ice and addition of 60 mM of MgCl$_2$. GTPγS-/GDP-loaded RAC1 and CDC42 as well as the native (WT) proteins were subjected to active RAC1/CDC42 pull-down assay using GST-PAK1-PBD fusion beads. Levels of GTP-loaded GTPases were detected using RAC1 or CDC42-specific antibodies.

### In vitro binding assay: Recombinant protein ELISA

Binding affinity of ERK3 protein to RAC1, CDC42, ARP2/3 complex and ARP3 protein was measured by ELISA. Purified RAC1, CDC42, and ARP3 proteins or ARP2/3 protein complex (Cytoskeleton) were used as bait and full-length GST-ERK3 (SignalChem) was used as a titrant protein. Purified GST protein was used as a negative control for ERK3 binding. Each reaction was ran in triplicates. 96-well Nunc MaxiSorp plates (Cat# 44-2404-21, Thermo Fisher Scientific) were coated with 100 ng/well of bait protein diluted in 100 µl of PBS pH 7.5 overnight at 4°C. Plates were washed with 300 µl/well of 0.5% Tween-20/PBS pH 7.5 and blocked with 300 µl of 1% BSA in PBS pH 7.5 for 1 hr at RT with gentle shaking. After washing with Tween-20/PBS, titrant proteins were added at 5, 10, 20, and 40 nM final concentration in 100 µl of 1% BSA/PBS and plates were incubated for 2 hr at RT with gentle shaking. Afterward, plates were washed with Tween-20/PBS and incubated with primary antibodies: anti-GST or anti-ERK3-specific antibody (1:500) in 100 µl/well of 1% BSA/PBS for 1 hr at RT, followed by washing with Tween-20/PBS and 1 hr incubation with HRP-conjugated goat anti-mouse IgG at 1:40,000 (Cat# A16066, Invitrogen) in 100 µl per well of Tween-20/PBS. After washing with Tween-20/PBS, plates were incubated with 100 ml/well of eBioscience TMB substrate solution (Cat# 00-4201-56, Invitrogen). Reactions were stopped with 50 µl of STOP solution (1 M H$_3$PO$_4$). Absorbance was measured at 450 nm with 570 nm reference using microplate reader Tecan SPARK. Multiple negative controls were used to establish each affinity assay, including coating protein + 1st Ab, coating protein + 2nd Ab, coating protein + 1st Ab + 2nd Ab, coating protein + titrant protein + 1st Ab only, coating protein + titrant protein + 2nd Ab only, coating protein + BSA + 1st Ab + 2nd Ab, coating with BSA + titrant protein + 1st Ab + 2nd Ab.

## GDP/GTP nucleotide exchange assays

GDP/GTP exchange assay was performed using recombinant RAC1 and CDC42 proteins in binding buffer containing: 20 mM Tris-HCl (pH 7.5), 100 mM NaCl, and protease/phosphatases inhibitors. Firstly, RhoGTPases were stripped of nucleotides by incubation with binding buffer containing 10 mM EDTA for 10 min at RT. Afterward, reactions were supplemented with 50 mM MgCl$_2$ and 500 µM GDP (Active RAC1/CDC42 pull-down kit, Thermo Fisher, Cat# 16118/19) and incubated for 15 min at 37°C with gentle agitation. Samples of GDP-loaded RhoGTPases were kept on ice for further analyses

as controls. The remaining GDP-loaded RAC1 and CDC42 were then incubated with 500 µM GTPγS (Thermo Fisher kit Cat# 16118/19) in the presence or absence of 2 nM of recombinant ERK3 protein (aa 9–327) (Crelux) for 30 min at 37°C with gentle agitation. All reactions were terminated by addition of 60 mM $MgCl_2$ on ice.

To isolate the active (GTP-bound) RAC1 and CDC42, the specific GST-PAK1-PBD-fusion beads were used. Nucleotide exchange reaction were incubated with the beads for 1 hr at 4°C, rotating. Afterward, beads were washed three times with binding/wash buffer: 25 mM Tris-HCl, pH 7.2, 150 mM NaCl, 5 mM $MgCl_2$, 1% NP-40, and 5% glycerol. Samples were eluted with 4× sample buffer supplemented with 100 mM DTT. Levels of active RAC1 and CDC42 were detected using RAC1 and CDC42-specific antibodies from the active RAC1/CDC42 pull-down kit (Cat# 16118/19, Thermo Fisher). Levels of GST-PAK1-PBD protein were detected using Ponceau S staining of the membrane. Efficiency of the GDP-GTP exchange was calculated as the ratio of GTPγS+ERK3 samples and GTPγS alone and presented as fold change where GTPγS samples were used as a reference.

Guanine nucleotide exchange (GEF) activity of ERK3 (9–327 aa) was further evaluated using RhoGEF Exchange Assay (Cat# BK100, Cytoskeleton). Assay was performed according to the manufacturer's instructions and included Dbs GEF as a positive control.

## In vitro kinase assay

In order to assess kinase activity of full-length (recombinant ERK3 Cat# M31-34G, SignalChem) and kinase domain ERK3 (ERK3 amino acids (aa) 9–327, Crelux), we performed in vitro kinase assay using recombinant MK5 (MAPKAPK5 Unactive Cat# M42-14G, SignalChem) and Myelin basic protein (MBP) as substrates (MBP, dephosphorylated, Cat# 13-110, Merck). For each reaction, 1 µg of the selected substrate (MK5 or MBP) was mixed with 0.25 µg of either full-length GST-ERK3, GST protein alone (*E. coli* expressed) or kinase domain ERK3 in kinase buffer (250 mM HEPES pH 7.5, 500 mM NaCl, 25 mM $MgCl_2$, supplemented with protease inhibitor cocktail Set I-Calbiochem 1:100; Cat# 539131, Merck Millipore) and phosphatase inhibitors: 1 mM sodium orthovanadate ($Na_3VO_4$), 1 mM sodium fluoride (NaF). The $MG^{2+}$/ATP-activating solution (Cat# BML-EW9805, Enzo) was added and reactions were incubated for 30 min at 30°C on a thermo-mixer with agitation at 550 rpm. All reactions were stopped by addition of 4× Laemmli buffer and boiled at 95°C for 5 min prior western blot analyses. For the MBP kinase assay, phospho-serine and phospho-threonine antibodies were used to assess MBP phosphorylation status. In case of MK5, specific phospho-MK (T182) antibody was used.

## In vivo F-actin/G-actin assay

HMECs and MDA-MB231 cells were seeded in 12-well or 6-well plates for F/G actin assay and parallel western blot analyses. When cells reached about 70% confluency, plates were subjected to either western blot analyses or G-actin/F-actin in vivo assay using cytoskeleton kit (Cat# BK037) according to the manufacturer's instructions. Briefly, cells were washed with 1× PBS and lysed in appropriate volume of lysis and F-actin stabilization buffer supplemented with 1 mM ATP and protease inhibitor mixture for F-actin stabilization (Cytoskeleton). Lysates were centrifuged at 100,000 × *g* at 37°C for 1 hr to separate the G-actin fraction in the supernatants and pelleted F-actin. F-actin pellets were further depolymerized for 1 hr on ice using F-actin depolymerizing buffer. Both sets of samples were mixed with 4× SDS-PAGE sample buffer. Equal volumes of G-actin and F-actin lysates were analyzed by SDS-PAGE and immunoblotting with anti-actin rabbit polyclonal antibody (Cat# AANO1, Cytoskeleton). Densitometric analyses of the G-actin and F-actin levels were performed using ImageJ software. Ratios of F-actin in respect to G-actin levels were calculated.

## In vitro actin polymerization assay

Actin polymerization assay was performed using actin polymerization biochem kit (Cat# BK003, Cytoskeleton) according to the provided protocol. Pyrene-labeled rabbit skeletal muscle actin (Cat# AP05-A, Cytoskeleton) (2.3 µM per reaction) was diluted with general actin buffer (5mM Tris-HCl pH 8.0, 0.2 mM $CaCl_2$) supplemented with 0.2 mM ATP and 1 mM DTT and incubated for 1 hr on ice to depolymerize actin oligomers. Actin stock was centrifuged for 30 min at 14,000 rpm at 4°C. Pyrene-labeled actin was incubated with recombinant ARP2/3 protein complex alone (Cat# RP01P, Cytoskeleton) or along with the human recombinant proteins: WASP-VCA domain protein (Cat# VCG03, Cytoskeleton) or ERK3 protein (M31-34G, SignalChem). ARP2/3 protein complex, WASP-VCA domain

protein, and full-length ERK3 recombinant protein were used at 10 nM, 400 nM, and 4.8 nM final concentration, respectively. Actin alone was used to establish a baseline of polymerization rate. Actin polymerization was induced by addition of 1.5× actin polymerization buffer (Cat# BSA02) (10× buffer: 20 mM MgCl$_2$, 500 mM KCl, 10 mM ATP, 50 mM guanidine carbonate in 100 mM Tris-HCl, pH 7.5). Actin polymerization was measured by fluorescence emission at 415 nm over 1 hr (30–60 s interval time) at RT with 360 nm excitation wavelength using multimode microplate reader Tecan SPARK. Buffer background signal at each interval was subtracted and relative fluorescence units (RFUs) are depicted.

## Immunofluorescence (IF) and confocal analyzes

Cells cultured on coverslips were fixed in 3.7% formaldehyde (Cat# CP10.1, Roth) for 15 min, followed by washing with PBS pH 7.5 and 3 min permeabilization using 0.1% Triton X-100 (AppliChem). After washing twice with PBS, cells were blocked with 1% BSA in PBS for 15 min and washed once with PBS. Filamentous actin was labeled with Oregon Green 488 Phalloidin (Cat# O7466, Invitrogen) or alternatively Rhodamine Phalloidin (Cat# R415, Invitrogen) in blocking buffer for 1 hr at RT in the dark. Nuclei were stained with 10 µg/ml of DNA dye (Hoechst 33342; Cat# H3570, Invitrogen). For co-staining of endogenous ERK3 and F-actin, cells were incubated in 1:250 dilution of anti-ERK3 antibody (mouse Cat# MAB3196, R&D or rabbit Cat# ab53277, Abcam) in blocking solution for 1 hr at RT. For the ARP3 and CDC42 staining, the anti-ARP3 antibody (Cat# ab151729, Abcam) and anti-CDC42 antibody (Cat# 610929, BD Transduction) were used in blocking buffer for 1 hr at RT in the dark. Nuclei were stained with 10 µg/ml of DNA dye (Hoechst 33342; Cat# H3570, Invitrogen). Afterward, cells were washed with PBS and incubated with secondary anti-mouse IgG-Cyanine3 (Cat# A10521, Thermo Fisher Scientific) at 5 µg/ml, DNA dye (Hoechst 33342), and Green Phalloidin in blocking solution for 1 hr at RT in the dark. Samples were washed twice with PBS and cells were mounted onto glass slides using Moviol (+DABCO) (Sigma). For exogenous ERK3 detection upon overexpression of the *ERK3* WT or serine 189 phosphorylation mutants (S189A/S189D) in sh*ERK3* HMECs, V5-tag-specific antibody was used at 1:100 dilution (Cat # R960-25, Invitrogen). Cells were imaged using a Leica DMi8 confocal microscope (×63, oil immersion objective). *ERK3*-depleted cells were used as a control for the ERK3 staining.

For the colocalization analyses, a mask was created to select excluding nuclei where no colocalization was observed. The ARP3 and ERK3 images were scaled and thresholded in the same way to maximize the dynamic scale and then had the colocalization of the Pearson's correlation and Spearman's rank correlation coefficients (PCC and SCC) on a pixel-wise basis with the 'Coloc 2' algorithm in ImageJ. For both the Pearson and Spearman correlation coefficients, a value of +1 is perfectly correlated or colocalized, a value of 0 means that the two signals are randomly localizing, and a value of −1 indicates that the two signals are perfectly separated.

## Scanning electron microscopy (SEM)

Control and ERK3-depleted HMECs were seeded in 12-well plate. Cells were fixed overnight with 2.5% glutaraldehyde, rinsed with PBS, and then again fixed and stained with 2% osmium tetroxide. Afterward, samples were rinsed with distilled water, frozen, and then freeze-dried with Crist Alpha LSC Plus Freeze Drier. The SEM scans were performed with a Philipps ESEM XL30 scanning electron microscope.

## Filopodia quantification and analyses

To detect, quantify, and analyze the number and lengths of the filopodia of the fixed and actin-phalloidin-stained control and *ERK3* knockdown HMECs, the FiloQuant open access software and routines (*Jacquemet et al., 2017*; *Jacquemet et al., 2019*) were applied in ImageJ and in some cases were applied in Imaris version 9.3.1 (Imaris, RRID:SCR_007370) for additional verification. Briefly, the raw cell images were linearly adjusted for brightness and contrast for optimal filopodial observation and also had a mask applied to just analyze the filopodial region of interest in each cell and applied to each cell in the same way. Additionally, the following parameters were initially applied to each cell in the same way: Cell Edge Threshold = 20, Number of Iterations = 10, Number of Erode Cycles = 0, Fill Holes on Edges = checked, Filopodia Threshold = 25, Filopodia Minimum Size = 10 pixel. The

results of the detected filopodial filaments were the overlayed in white over the actin-phalloidin green fluorescence of the filapodia for visual verification in ImageJ and in some cases in Imaris version 9.3.1.

## Live-cell imaging and cell tracking analyses

GFP-actin expressing control and ERK3 knockdown MDA-MB231 cells were cultured in sterile chambers (1 × 10^4 cells/well) (μ-slide 8-well, ibidi), coated with 100 μl/well of 5% collagen in DMEM + FBS (Rat tail collagen, Cat# 5056, Advanced BioMatrix) and kept under 5% $CO_2$, 37 °C, and 90% humidity in a OkoLabs environmental incubator (H-301K environmental chamber, Oko Touch, Oko Pump, T-Control, and CO2 control, OkoLabs) on the Leica Confocal microscope table.

The images of both GFP fluorescence and transmission of the live cells were acquired with a Leica SP8 confocal microscope using a 10 × 0.3 NA objective, with 488 nm excitation (at approximately 150 μW) and emission window of 500–590 nm for GFP detection and with scanning differential interference contrast transmission imaging in a 1500 μm × 1500 μm frame format with 400 lines per second, 0.71 μm/pixel (2048 × 2048 pixel per frame), and with two times averaging per line with a frame acquisition of every 30 min per selected position within the chamber. The images were first acquired in multiple regions of each cell type for 20 h.

The image sequences were imported into Imaris version 9.3.1 and detected automatically by fluorescence with both the whole-cell spot and whole-cell surface analysis. As the surface and spot analysis center positions differed in control analysis by less than 1%, the whole cell spot automated analysis was applied to all images in the same way with a 16 μm per cell diameter estimate. The automated tracking also occurred in the same way for all image sequences within Imaris with the autoregressive motion algorithm, with a maximum average distance of 60 μm per step and with zero step gap applied. The speed (or cell speed) was determined by dividing each x, y step by 1800s (or 30 min). The acceleration (or cell acceleration) was determined by subtracting a step speed from the previous step speed and dividing by 1800s (or 30 min). The displacement length was determined by subtracting the initial x, y position from the final x, y position and then to determine the difference vector length for each track over 20 hr of acquisition. The track length added each absolute x, y vector step for an entire track over 20 hr of acquisition. The track mean speed averaged the speed of the steps for individual tracks.

The graphs were created with GraphPad Prism (RRID:SCR_002798), and cell distribution according to the analyzed parameter was visualized by violin plots. Significance was determined using non-parametric Mann–Whitney test.

## Transwell cell migration assay

Migratory properties of control and ERK3-depleted HMECs and MDA-MB231 cells were assessed using two-chamber Transwell system (Cat# 3422, Corning). Cells were seeded in 12-well plates. HMECs were deprived of supplements from the medium 24 hr prior the migration. HMECs and MDA-MB231 cells were then trypsinized and resuspended in supplements-free or serum-free medium, respectively. 500 μl of supplements-free/serum-free medium was mixed with 100 ng/ml of EGF and added into the bottom chamber and 1 × 10^5 cells in 120 μl of supplements-free/serum-free medium were added into each insert. Plates were incubated at 37°C for 24 hr. To quantify the migrated cells, cells were removed from the upper surface of the insert using cotton swabs and inserts were washed with PBS. Afterward, cells that migrated to the lower side were fixed within 3.7% formaldehyde for 10 min and stained with Hoechst solution in PBS for 20 min at 37°C. Cells were then visualized using Leica DMi8 microscope (×5 dry objective), images of 3–4 regions per each experimental condition were taken. Number of cells was quantified by Fiji/ImageJ software (Fiji, RRID:SCR_002285) using particle analyses for each field of view and averaged for each membrane. Directional migration was presented and the percentage of the respective control (expressed as 100%).

## Phosphorylation site identification on ARP3

Phosphorylation of the ARP2/3 protein complex subunits by full-length ERK3 was detected by in vitro kinase assay and in-gel digestion of the respective ARP2/3 complex proteins followed by subsequent spectrometry analyses.

## In-gel digestion

The protein in the gel lanes were digested with 0.1 µg trypsin (Promega) in 20 µl 25 mM ammonium bicarbonate, pH 7.8 at 37°C for 16 hr. The tryptic peptides were purified OMIX C18, 10 µl SPE tips (Agilent, Santa Clara, CA), and dried using a Speed Vac concentrator (Savant, Holbrook, NY).

## Phosphopeptide enrichment

Dried tryptic peptide samples were dissolved in loading buffer (1 M glycolic acid, 6% trifluoroacetic acid, 5% glycerol, and 80% acetonitrile) under continuous shaking. $TiO_2$ beads (Titansphere, $TiO_2$, GL Sciences Inc) were washed in loading buffer three times before transferring them to the dissolved tryptic peptide samples. After 1 hr of continuous shaking, the supernatant was collected and transferred to a new tube containing freshly washed $TiO_2$ beads for a second incubation. The $TiO_2$ beads were collected separately and gently washed with 200 µl of loading buffer, 200 µl 80% acetonitrile/2% trifluoroacetic acid, 200 mM ammonium glutamate, and 200 µl of 50% acetonitrile/1% trifluoroacetic acid, respectively. The $TiO_2$ beads were dried and bound peptides were eluted sequentially in 10 min at first with 50 µl of 10% ammonium hydroxide, pH 11.7, then with 50 µl of 15% ammonium hydroxide/60% acetonitrile, and finally with 50 µl of 1% pyrrolidine. Eluted peptides were acidified by adding 75 µl 50% formic acid and cleaned up using OMIX C18, 10 µl SPE tips (Agilent).

## LC-MS analysis

The samples were solved in 10 µl 0.1% formic acid and 5 µl was analyzed by LC-MS using a timsTOF Pro (Bruker Daltonik, Bremen, Germany), which was coupled online to a nanoElute nanoflow liquid chromatography system (Bruker Daltonik) via a CaptiveSpray nanoelectrospray ion source. The peptides were separated on a reversed-phase C18 column (25 cm × 75 µm, 1.6 µm, IonOpticks; Fitzroy, VIC, Australia). Mobile phase A contained water with 0.1% (vol/vol) formic acid, and acetonitrile with 0.1% (vol/vol) formic acid was used as mobile phase B. The peptides were separated by a gradient from 0 to 35% mobile phase B over 25 min at a flow rate of 300 nl/min at a column temperature of 50°C. MS acquisition was performed in DDA-PASEF mode.

## ERK3-dependent tumor cell motility in vivo

All in vivo experiments were performed in accordance with the Swiss animal welfare ordinance and approved by the cantonal veterinary office Basel-Stadt. Female NSG mice were maintained in the Department of Biomedicine animal facilities in accordance with Swiss guidelines on animal experimentation (license number 2464). NSG mice are from in-house colonies. Mice were maintained in a sterile-controlled environment (a gradual light–dark cycle with light from 7:00 to 17:00, 21–25°C, 45–65% humidity).

For engraftment of MDA-MB231-GFP, $0.5 \times 10^6$ cells were suspended in 50 µl Matrigel in PBS (1:1) and injected into the fourth mouse mammary gland of an 8-wk-old female. Orthotopic mammary tumors were grown for 4–5 wk.

## Intravital imaging

Mice were anesthetized with Attane Isofluran (Provet AG) and anesthesia was maintained throughout the experiment with a nose cone. Tumors were exposed by skin flap surgery on a Nikon Ti2 A1plus multiphoton microscope and imaged at 880 nm with an Apochromat ×25/1.1 NA water immersion objective at a resolution of 1.058 µm per pixel. Cell motility was monitored by time-lapse imaging over 30 min in 2 min cycles, where a 100 mm Z-stack at 5-mm increments was recorded for each field of view starting at the tumor capsule. Three-dimensional time-lapse videos were analyzed using ImageJ. Images were registered using demon algorithm in MATLAB to correct for breathing movement (*Kroon, 2023*). Tumor cell motility was quantified manually. A tumor cell motility event was defined as a protrusion of half a cell length or more over the course of a 30 min video.

## Statistical analyses

All experiments were repeated at least three times and exact replicate number (n) is specified for each figure. GraphPad Prism 9 was used to analyze data. Analyses performed for each data set, including statistical test and post-test, were specified for each figure. Where applicable, data are presented

as mean ± SEM of at least three independent experiments. Significance levels are displayed in GP (GraphPad) style: *p<0.0332, **p<0.0021, ***p<0.0002, ****p<0.0001.

## Acknowledgements

The authors thank Kerstin Bahr, Institute of Functional and Clinical Anatomy, University Medical Center of the Johannes Gutenberg University, Mainz, for electron microscopy imaging. We would like to thank Stefanie Wenzel and Lisa Winkler for the technical assistance. This work was supported by the grants from Else Kroener Fresenius Stiftung; MERCK (project ID-ERK-KR). We would like to thank the Translational oncology team of MERCK for their valuable inputs. We would like to thank Dr. Raphael Thierry and Dr. Ewelina Bartoszek for their assistance with MATLAB script used for the analyses of the tumor cell motility analyses. We would like to thank Dr. Ulrike Theisen and Prof. Reinhard W Köster from the department of Cellular and Molecular Neurobiology, of Technische Universität Braunschweig, Braunschweig, Germany, for critical reading of the mansucript and valuable input. We thank Dr. Daniela Hoeller for the critical reading and editing of this manuscript.

## Additional information

### Competing interests

Krishnaraj Rajalingam: KR is the founder and MD of KHR Biotec GmbH. The other authors declare that no competing interests exist.

### Funding

| Funder | Grant reference number | Author |
| --- | --- | --- |
| Else Kröner-Fresenius-Stiftung | SUN-MAPK | Katarzyna Bogucka-Janczi Gregory Harms |
| Deutsche Forschungsgemeinschaft | CRC1292/TP05 | Katarzyna Bogucka-Janczi Krishnaraj Rajalingam |

The funders had no role in study design, data collection and interpretation, or the decision to submit the work for publication.

### Author contributions

Katarzyna Bogucka-Janczi, Conceptualization, Data curation, Formal analysis, Validation, Investigation, Visualization, Writing - original draft, Project administration, Writing – review and editing; Gregory Harms, Software, Formal analysis, Methodology; Marie-May Coissieux, Data curation, Formal analysis, Investigation, Visualization, Methodology, Writing – review and editing; Mohamed Bentires-Alj, Conceptualization, Formal analysis, Supervision, Investigation, Methodology, Project administration; Bernd Thiede, Data curation, Software, Formal analysis, Validation, Investigation, Methodology; Krishnaraj Rajalingam, Conceptualization, Formal analysis, Supervision, Funding acquisition, Investigation, Writing - original draft, Project administration, Writing – review and editing

### Author ORCIDs

Katarzyna Bogucka-Janczi http://orcid.org/0000-0001-6254-3359
Marie-May Coissieux http://orcid.org/0000-0001-5017-5253
Krishnaraj Rajalingam http://orcid.org/0000-0002-4175-9633

### Ethics

The animal experiments were performed as per the guidelines of University Medical Center Basel.

### Decision letter and Author response

Decision letter https://doi.org/10.7554/eLife.85167.sa1
Author response https://doi.org/10.7554/eLife.85167.sa2

## Additional files

### Supplementary files

• MDAR checklist

• Supplementary file 1. Excel file with the identified phosphopeptides of ARP3. Phosphorylation of the ARP2/3 protein complex subunits by full-length ERK3 was detected by in vitro kinase assay and in-gel digestion of the respective ARP2/3 complex proteins followed by subsequent LC-MS analysis using a timsTOF Pro coupled to a nanoElute nanoflow liquid chromatography system. MS acquisition was performed in DDA-PASEF mode.

### Data availability

All source data files have been uploaded with the manuscript.

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

# Appendix 1

## Appendix 1—key resources table

| Reagent type (species) or resource | Designation | Source or reference | Identifiers | Additional information |
|---|---|---|---|---|
| Cell line (*Homo sapiens*) | Immortalized human mammary epithelial cells hTERT-HME1 (ME16C) (HMECs) | ATCC | ATCC CRL-4010 | |
| Cell line (*H. sapiens*) | MDA-MB231 | DSMZ | Cat# ACC 732 | |
| Cell line (*H. sapiens*) | MDA-MB231-GFP | Gift from Prof. Dr. Mohamed Bentires-Alj (University of Basel, Department of Biomedicine) | | |
| Cell line (*H. sapiens*) | 293T cells | Gift from Dr. Andreas Ernst (Goethe-University Frankfurt am Main, IBC2) | | |
| Sequence-based reagent-shRNA | shRNA: *ERK3 (MAPK6)* NM_002748.x-3734s1c1 CCGG GCTGTCCACGTACTTAATTTACTC GAGTAAATTAAGTACGTGGA CAGCTTTTT | MISSION shRNA Human Library (Sigma) | TRCN0000001568 | |
| Sequence-based reagent-shRNA | sh*CDC42*#1 NM_001791.2-471s1c1 | MISSION shRNA Human Library (Sigma) | TRCN0000047628 | Sequence: CCGGCCCTCTACTATTGAGA AACTTCTCGAGAAGTTTCTC AATAGTAGAGGGTTTTTG |
| Sequence-based reagent-shRNA | sh*CDC42*CDC42#2 NM_001791.2-193s1c1 | MISSION shRNA Human Library (Sigma) | TRCN0000047629 | Sequence: CCGGCGGAATATGTACCGAC TGTTTCTCGAGAAACAGTCG GTACATATTCCGTTTTTG |
| Sequence-based reagent-siRNA | siRNA: *ERK3 (MAPK6)*, Hs_MAPK6_5: 5′- AGUUCAAUUUGA AAGGAAATT-3′ | QIAGEN | Cat# SI00606025 | |
| Sequence-based reagent-CRISPR/Cas9 | CrispR/Cas gRNA sequences targeting *ERK3 (MAPK6)* (CrispR ERK3) | Sigma, *Doench et al., 2016* | Designed by Rule Set 2 of Azimuth 2.0 as described previously | #1 5′-CACCGAGCCAATTAAC AGACGATGT-3′ #2 5′-CACCGATACTTGTAACTACA AAACG-3′ #3 5′-CACCGCTGCTGTTAAC CGATCCATG-3′ |
| Recombinant DNA reagent-cDNA | *ERK3* K49A K50A kinase dead mutant | Site-directed mutagenesis | | Primers sequence: frw_5′ GCAATTGTCCTTACTG ATCCCCAGAGTGTC, rev_5′ CGCGATGGCTACTCTTTTGT CACAGTC |
| Recombinant DNA reagent-cDNA | *ARP3* pENTR223 | Harvard | HsCD00375598 | |
| Recombinant DNA reagent-cDNA | *ARP3* S418A | Eurofins | Cat# #11104144057-1-5 | |
| Recombinant DNA reagent-cDNA | *ARP3* S418D | Eurofins | Cat# #11104144057-1-6 | |
| Recombinant DNA reagent-cDNA | pGEX-2T-*RAC1*-WT | Addgene | Addgene plasmid# 12977 | |
| Recombinant DNA reagent-cDNA | pGEX-2T-*CDC42*-WT | Addgene | Addgene plasmid #12969 | |
| Recombinant DNA reagent-cDNA | pGEX2TK-*PAK1* (70–117) containing the p21-binding domain (PBD) | Addgene | Addgene plasmid# 12217 | |
| Sequence-based reagent | qRT-PCR primers, human *ERK3 (MAPK6)* | Sigma | | Frw_5′ ATGGATGAGCCAATTTCAAG, Rv_5′ CTGACAATCATGATAC CTTTCC |
| Sequence-based reagent | qRT-PCR primers, human *18S* | Sigma | Cat# 88-8086 | Frw_5′ AGAAACGGCTACCACATCCA, Rv_5′ CACCAGACTTGCCCTCCA |
| Commercial assay or kit | Cell Fractionation Kit | Invent Biotechnologies | Cat# SM-005 | |

*Appendix 1 Continued on next page*

*Appendix 1 Continued*

| Reagent type (species) or resource | Designation | Source or reference | Identifiers | Additional information |
|---|---|---|---|---|
| Commercial assay or kit | RAC1 pull-down and detection kit | Thermo Fisher Scientific | Cat# 16118 | |
| Commercial assay or kit | CDC42 pull-down and detection kit | Thermo Fisher Scientific | Cat# 16119 | |
| Commercial assay or kit | TNT T7 Quick Coupled transcription/Translation System | Promega | Cat# TM045 | |
| Sequence-based reagent | qRT-PCR primers, human *ERK3 (MAPK6)* | Sigma | | Frw_5' ATGGATGAGCCAATTTCAAG, Rv_5' CTGACAATCATGATAC CTTTCC |
| Sequence-based reagent | qRT-PCR primers, human *18S* | Sigma | Cat# 88-8086 | Frw_5' AGAAACGGCTACCACATCCA, Rv_5' CACCAGACTTGCCCTCCA |
| Commercial assay or kit | Cell Fractionation Kit | Invent Biotechnologies | Cat# SM-005 | |
| Commercial assay or kit | RAC1 pull-down and detection kit | Thermo Fisher Scientific | Cat# 16118 | |
| Commercial assay or kit | CDC42 pull-down and detection kit | Thermo Fisher Scientific | Cat# 16119 | |
| Commercial assay or kit | TNT T7 Quick Coupled transcription/Translation System | Promega | Cat# TM045 | |
| Commercial assay or kit | RhoGEF Exchange Assay | Cytoskeleton | Cat# BK100 | |
| Commercial assay or kit | G-actin/F-actin in vivo assay | Cytoskeleton | Cat# BK037 | |
| Commercial assay or kit | Actin polymerization biochem kit | Cytoskeleton | Cat# BK003 | |
| Commercial assay or kit | Two-chamber transwell system | Corning | Cat# 3422 | |
| Antibody | Anti-beta-Actin HRP conjugated (mouse monoclonal) | Abcam | Cat# ab49900 | WB 1:40,000 |
| Antibody | Anti-GAPDH antibody (mouse monoclonal) | GeneTex | Cat# GTX627408 | WB 1:1000 |
| Antibody | HRP-conjugated secondary antibody for rabbit IgG (goat) | Invitrogen | Cat# A16096 | WB 1:40,000 |
| Antibody | HRP-conjugated secondary antibody for rabbit IgG (goat) | Invitrogen | Cat# 32460 | WB 1:2000 |
| Antibody | HRP-conjugated secondary antibody for mouse IgG (sheep) | GE Healthcare Life Sciences | Cat# NA9310 | WB 1:20,000 |
| Antibody | Anti-human ERK3 (mouse monoclonal) | R&D | Cat# MAB3196 | IF 1:400 |
| Antibody | Anti-phospho-ERK3 (pSer189) (rabbit polyclonal) | Sigma | Cat# SAB4504175 | 1:500 |
| Antibody | Anti-ARPC1A (rabbit polyclonal) | Sigma | Cat# HPA004334 | 1:500 |
| Antibody | Anti-phospho-p44/42 MAPK (ERK1/2) (Thr202/Tyr204) (rabbit polyclonal) | Cell Signaling Technology | Cat# 9101L | WB 1:1000 |
| Antibody | Anti-V5 Tag (mouse monoclonal) | Invitrogen | Cat # R960-25 | 1:1000 |
| Antibody | Anti-GST (rabbit polyclonal) | Cell Signaling Technology | Cat# 2622S | 1:1000 |
| Antibody | Anti-normal rabbit IgG (rabbit polyclonal) | Cell Signaling Technology | Cat# 2729 | Used as a control for IP |
| Antibody | Anti-GST (B-14) (mouse monoclonal) | Santa Cruz Biotechnology | Cat# sc-138 | WB 1:1000 |
| Antibody | Anti-RAC1 (mouse monoclonal) | BD Transduction Laboratories | Cat# 610651 | WB 1:1000 |
| Antibody | Anti-CDC42 (mouse monoclonal) | BD Transduction Laboratories | Cat# 610929 | WB 1:1000 |
| Antibody | ARP3 (rabbit monoclonal) | Abcam | Cat# ab151729 | WB 1:1000 |
| Antibody | ARP2 (rabbit monoclonal) | Abcam | Cat# ab128934 | WB 1:1000 |
| Antibody | Anti-ERK3 (mouse monoclonal) | R&D | Cat# MAB3196 | 1:250 |
| Antibody | Anti-rabbit IgG-Alexa 488 (goat) | Thermo Fisher Scientific | Cat# A11008 | IF, working concentration: 5 µg/ml |

*Appendix 1 Continued on next page*

*Appendix 1 Continued*

| Reagent type (species) or resource | Designation | Source or reference | Identifiers | Additional information |
|---|---|---|---|---|
| Antibody | Anti-mouse IgG-Cyanine3 (goat) | Thermo Fisher Scientific | Cat# A10521 | IF, working concentration:: 5 µg/ml |
| Chemical compound, drug | Hoechst 33342 | Invitrogen | Cat# H3570 | 10 µg/ml |
| Chemical compound, drug | Oregon Green 488 Phalloidin | Invitrogen | Cat# O7466 | 1:50 |
| Chemical compound, drug | Human EGF recombinant protein (human) | Invitrogen | Cat# RP-10927 | 100 ng/ml |
| Chemical compound, drug | Cycloheximide (CHX) | Sigma | Cat# C-7698 | 100 µg/ml |
| Peptide, recombinant protein | ERK3 (MAPK6) kinase domain (aa 9–327) (human) | Crelux | Customized | |
| Peptide, recombinant protein | ERK3 (MAPK6) recombinant protein (human) | Aviva Systems Biology | Cat# OPCA01714 | |
| Peptide, recombinant protein | ERK3 (MAPK6) protein full-length (human) | SignalChem | Cat# M31 | |
| Peptide, recombinant protein | WASP Protein VCA Domain (human) | Cytoskeleton | Cat# VCG03 | |
| Peptide, recombinant protein | ARP2/3 Protein Complex: Porcine Brain | Cytoskeleton | Cat# RP01P | |
| Software, algorithm | ImageJ | RRID:SCR_003070 | RRID:SCR_003070 https://imagej.net/ | |
| Software, algorithm | ImageJ Coloc2 Plugin | Self-modified version as described by *French et al., 2008*. | Self-modified version of ImageJ; RRID:SCR_003070 | |
| Software, algorithm | Fiji | RRID:SCR_003070 | Fiji (RRID:SCR_002285) | |
| Software, algorithm | Imaris version 9.3.1 | RRID:SCR_003070 | Imaris, RRID: SCR_007370 | |
| Software, algorithm | ImageJ | RRID:SCR_003070 | RRID:SCR_003070 https://imagej.net/ | |
| Software, algorithm | ImageJ Coloc2 Plugin | Self-modified version as described by *French et al., 2008*. | Self-modified version of ImageJ; RRID:SCR_003070 | |
| Software, algorithm | Fiji | RRID:SCR_003070 | Fiji (RRID:SCR_002285) | |

