## [Editor Report]

The manuscript describes a fundamental study of the atypical MAPK, ERK3, in the activation of RhoGTPase Cdc42 and the formation of actin-rich protrusions and cell migration. The results based on compelling evidence show that ERK3 is required for the motility of tumor cells in vivo, providing a new target for fighting metastasis.

---

## [Decision Letter]

**Decision letter after peer review:**

Thank you for submitting your article "ERK3/MAPK6 dictates Cdc42/Rac1 activity and ARP2/3-dependent actin polymerization" for consideration by *eLife*. Your article has been reviewed by 3 peer reviewers, and the evaluation has been overseen by a Reviewing Editor and Volker Dötsch as the Senior Editor. The reviewers have opted to remain anonymous.

Essential revisions:

1. One of the key discoveries is that ERK3 is functioning as a direct GEF for Cdc42 but not Rac1. Using cell-based studies, the authors performed reconstitution experiments with wildtype and kinase-dead versions of ERK3 under ERK3 depleted conditions to delineate the role of the kinase activity of ERK3 in mediating Rac1/Cdc42 activity. However, it would be helpful to know if the authors also tried to reconstitute cells with the kinase domain alone and check for effects on Cdc42.

2. Related: The authors' efforts to show and understand the effects of the ERK3:small GTPase and ERK3:ARP3 interactions are appreciated. They show that the kinase domain is sufficient to bind Rac1 and Cdc42 (Figure 2L). For ARP3, binding with full-length ERK3 is shown (Figure 6C). Is the ERK3 kinase domain also sufficient for ARP3 binding?

3. The authors employ purified ERK3 protein (both full-length and kinase domain) to perform direct GEF assays. Is there any difference in the activity of the proteins employed as they seem to be obtained from different sources? It may also be interesting to test the activity of these proteins in a direct kinase assay.

4. The data demonstrating the dependency of ERK3 for Arp3S418 phosphorylation in cells is hidden in the supplement. This should be presented in the main figure. Details of the ARP3 S418 antibody need to be provided.

5. It is not clear which isoform of DBS has been used for the Arp2/3 polymerization assays. Please clarify.

6. The results presented in 1J and the supporting movies are very convincing. Have these effects been investigated on metastatic seeding with these cells?

7. Figure 6: Supplement 2- These observations also demonstrate that there is a possible feedback working on ERK3 -mediated Cdc42 activation as Cdc42 activity is reverted in cells expressing Arp3S418D but not the F actin content or chemotactic migration of cells. Do they exhibit actin-rich protrusions?

8. Do the authors see any effect with the knockdown of ERK4 as they are closely related?

9. ERK3 is activated by PAK kinases (through phosphorylation at Ser 189) which are downstream of Rac1 and Cdc42 and the authors can add further discussion in these lines in the context of the summary figure presented in figure 8.

10. Figure 7 is rather important and could be better utilised to bring clarity to their investigation. Moving the Rac1/Cdc42 activity data from Figure 7 -suppl2 would add important functional information. This should be complemented with a blot for pARP3-S418 in the expression of empty vector/ERK3 WT/ERK3 KD conditions. Together this would strengthen the authors' conclusions in the text.

11. In Figure 3F the authors show a convincing GEF assay with the ERK3 kinase domain and Cdc42. The experiment shown in Figure supp 1 has a very high starting fluorescence, can the authors explain what is going on here?

12. In Figure 4, the colocalization data is supportive, but there are no real controls. The authors should consider changing their wording on line 280 in the results text, from "corroborated" to "supported".

13. It seems that none, or very little, of the overexpressed ERK3 (WT or mutant), is located in the vicinity of the actin cytoskeleton at the periphery of the cell. The authors should comment on this.

*Reviewer #1 (Recommendations for the authors):*

Below are some questions and suggestions for the authors.

1. One of the key discoveries is that ERK3 is functioning as a direct GEF for Cdc42 but not Rac1. Using cell-based studies, the authors performed reconstitution experiments with wildtype and kinase-dead versions of ERK3 under ERK3 depleted conditions to delineate the role of the kinase activity of ERK3 in mediating Rac1/Cdc42 activity. However, it would be helpful to know if the authors also tried to reconstitute cells with the kinase domain alone and check for effects on Cdc42.

2. The authors employ purified ERK3 protein (both full-length and kinase domain) to perform direct GEF assays. Is there any difference in the activity of the proteins employed as they seem to be obtained from different sources? It may also be interesting to test the activity of these proteins in a direct kinase assay.

3. Details of the ARP3 S418 antibody need to be provided.

4. It is not clear which isoform of DBS has been used for the Arp2/3 polymerization assays. Please clarify.

5. The results presented in 1J and the supporting movies are very convincing. However, I am curious if the authors investigated the effects of metastatic seeding with these cells.

*Reviewer #2 (Recommendations for the authors):*

Below are some questions and suggestions for the authors.

1. Apparently the kinase domain of ERK3 is enough to activate Cdc42 directly. I would propose the authors test the kinase activity of the different proteins they have employed here. Are they phosphorylated at Ser 189?

2. The data demonstrating the dependency of ERK3 for Arp3S418 phosphorylation in cells is hidden in the supplement. I would present this in the main figure. Also is this antibody commercially available?

3. Figure 6: Supplement 2- These observations also demonstrate that there is a possible feedback working on ERK3 -mediated Cdc42 activation as Cdc42 activity is reverted in cells expressing Arp3S418D but not the F actin content or chemotactic migration of cells. Do they exhibit actin-rich protrusions?

4. Do the authors see any effect with the knockdown of ERK4 as they are closely related?

5. ERK3 is activated by PAK kinases (through phosphorylation at Ser 189) which are downstream of Rac1 and Cdc42 and the authors can add further discussion in these lines in the context of the summary figure presented in Figure 8.

*Reviewer #3 (Recommendations for the authors):*

After reading the manuscript a number of questions arise that the authors could address to strengthen the manuscript.

1. The paper is heavily centered on ERK3 knock-down, which is acceptable and the authors have done their best to address this in multiple cell lines. However, the data shown in Figure 1-supp3 is concerning. Firstly, the authors should show the single V5-tag channel as it is not possible to critically comment on this as shown. Having said that, it seems that none, or very little, of the overexpressed ERK3 (WT or mutant), is located in the vicinity of the actin cytoskeleton at the periphery of the cell. The authors should comment on this. How does this experiment look if the anti-ERK3 antibody used in Figure 1A is employed here?

2. Related to the point above. When considering Figure 1, the first experiment that comes to mind is whether the effects seen in Figure 1A on the actin cytoskeleton can be rescued by reintroducing siRNA-insensitive ERK3. The authors do this experiment and come back to it in Figure 7 from a different angle. It would be informative to know what small GTPase activity/pARP3-S418 phosphorylation looks like in the ERK3 WT/S189A/S189D overexpression experiment.

3. In Figure 3F the authors show a convincing GEF assay with the ERK3 kinase domain and Cdc42. The experiment shown in Figure supp 1 has a very high starting fluorescence, can the authors explain what is going on here?

4. In Figure 4, the colocalization data is supportive, but there are no real controls. The authors should consider changing their wording on line 280 in the results text, from "corroborated" to "supported".

5. The authors' efforts to show and understand the effects of the ERK3:small GTPase and ERK3:ARP3 interactions are appreciated. They show that the kinase domain is sufficient to bind Rac1 and Cdc42 (Figure 2L). For ARP3, binding with full-length ERK3 is shown (Figure 6C). Is the ERK3 kinase domain also sufficient for ARP3 binding? A natural question is whether ERK3 binds ARP3 and Cdc42 at the same time, or would this be mutually exclusive. Can the authors address this with any data?

6. Figure 7 is rather important and could be better utilised to bring clarity to their investigation. Moving the Rac1/Cdc42 activity data from Figure 7 -suppl2 would add important functional information. This should be complemented with a blot for pARP3-S418 in the expression of empty vector/ERK3 WT/ERK3 KD conditions. Together this would strengthen the authors' conclusions in the text.

7. More of a comment. The authors convincingly show that EGF stim leads to pERK3-S189 phosphorylation and that ERK3 KD affects the EGF-dependent effects on the cytoskeleton. In spite of their efforts, it still remains rather unclear how this impacts the pERK3-small GTPase dynamics.

---

## [Author Response]

Essential revisions:1. One of the key discoveries is that ERK3 is functioning as a direct GEF for Cdc42 but not Rac1. Using cell-based studies, the authors performed reconstitution experiments with wildtype and kinase-dead versions of ERK3 under ERK3 depleted conditions to delineate the role of the kinase activity of ERK3 in mediating Rac1/Cdc42 activity. However, it would be helpful to know if the authors also tried to reconstitute cells with the kinase domain alone and check for effects on Cdc42.

Thanks for this comment. We have attempted to perform the active Rac1 and Cdc42 pulldowns from the cells overexpressing the kinase domain ERK3 vs. full-length ERK3 under conditions where ERK3 is depleted with a 3’UTR shRNA. However, due to the uneven expression levels of the truncated ERK3 construct (kinase domain aa 1-340) as compared to the WT ERK3, we were unable to reliably compare the active Rac1/Cdc42 pull-down from the cells overexpressing both constructs (please see Author response image 1 presented for the reviewer). Also please note that reconstruction of full-length ERK3 in ERK3 knockdown cells consistently brings back active Cdc42 levels (see also Figure 7E), while the expression of the kinase-dead (KD) version (ERK3 K49A/K50A in the ATP-binding site of ERK3) under the same settings failed to do so.

**Author response image 1. sa2fig1:** Western Blot analyses are presented to demonstrate the different expression efficiency of ERK3 WT (full-length) and truncated ERK3 (aa 1-340, kinase domain only) constructs with either N-Tap Flag tag (left panel) or C-Tap V5-tag (right panel). 1 µg cDNA was used for the transfection in 6-well plates as described in the methods section. Western Blot analyses show the total cell lysates used for the active Rac1/Cdc42 pull-down on the same membranes using two different antibodies (anti-Flag or anQ-V5, respectively).

2. Related: The authors' efforts to show and understand the effects of the ERK3:small GTPase and ERK3:ARP3 interactions are appreciated. They show that the kinase domain is sufficient to bind Rac1 and Cdc42 (Figure 2L). For ARP3, binding with full-length ERK3 is shown (Figure 6C). Is the ERK3 kinase domain also sufficient for ARP3 binding?

We found that the ERK3 kinase domain binds directly to the Arp3 protein, albeit with low affinity in comparison to the full-length protein. These data are now presented in the supplement (please see Figure 6, Supplement 1 A). The main text has been modified appropriately.

3. The authors employ purified ERK3 protein (both full-length and kinase domain) to perform direct GEF assays. Is there any difference in the activity of the proteins employed as they seem to be obtained from different sources? It may also be interesting to test the activity of these proteins in a direct kinase assay.

Very good question indeed. The full-length ERK3 (Cat# M31-34G SignalChem) was purified from Sf9 cells while the kinase domain of ERK3 (aa 9-327) (Crelux, custom order) was purified from *E. coli*. Our MassSpec analyses with both proteins confirmed that the kinase domain of ERK3 purified from *E. coli* lacks the S189 phosphorylation as compared to the full-length ERK3 purified from Sf9 cells. In Author response image 2 we present a screen shot of the MassSpec analyses using the two recombinant ERK3 proteins (Author response image 2). Peptides aa 180190 are presented here for the eyes of the reviewer with the phosphorylation at S189 indicated by the arrow. Identified peptides are marked in yellow, while the modifications are indicated in green (Scaffold software).

Moreover, these results are also evident in our in vitro kinase assays: The *E. coli*-purified kinase domain of ERK3 does not phosphorylate T182 of MK5, a known substrate of ERK3 as compared to the full-length ERK3 purified from Sf9 cells. Please see Figure 3 figure supplement 1B. We additionally employed MBP as a substrate to verify the serine and threonine phosphorylations and reached the same conclusions.

**Author response image 2. sa2fig2:** ERK3 peptides (aa 180-190) are presented from the MassSpec analyses using the recombinant kinase domain of ERK3 (top panel) or full-length ERK3 (bottom panel). Arrows indicate S189. Yellow marking indicates the identified peptide, and post-translational modifications are marked in green (Scaffold).

4. The data demonstrating the dependency of ERK3 for Arp3S418 phosphorylation in cells is hidden in the supplement. This should be presented in the main figure. Details of the ARP3 S418 antibody need to be provided.

Thanks for this suggestion. We have moved this data to the main figures as suggested (please see Figure 6H). The details of the antibody have been provided in the Materials and methods section.

5. It is not clear which isoform of DBS has been used for the Arp2/3 polymerization assays. Please clarify.

We used DBS provided by the manufacturer (Cytoskeleton.inc). DBS is known to exert strong GEF activity towards Cdc42 and RhoA, but is not the most potent GEF for Rac1 in vitro^16,17^. Moreover, we used this protein as a positive control for Cdc42 and as shown in Figure 3F, it functioned according to its described function.

The isoform information for DBS protein (Cat# GE1-A; Cytoskeleton.inc) was not specified in the manual. However, we contacted the manufacturer and obtain the following information: “Highlighted in green is the DH/PH domain that is in GE01. We believe it is isoform #4.”

**Author response image 3. sa2fig3:** 

6. The results presented in 1J and the supporting movies are very convincing. Have these effects been investigated on metastatic seeding with these cells?

Good suggestion. These experiments have been presented in our previous work (Bogucka et al., *eLife* 2020) and demonstrated that ERK3 is required for the metastatic seeding of these cells.

7. Figure 6: Supplement 2- These observations also demonstrate that there is a possible feedback working on ERK3 -mediated Cdc42 activation as Cdc42 activity is reverted in cells expressing Arp3S418D but not the F actin content or chemotactic migration of cells. Do they exhibit actin-rich protrusions?

Its indeed intriguing that Cdc42 but not Rac1 activity is reverted in ArpS418D expressing cells despite the KD of ERK3. We need to perform additional studies to delineate the mechanisms behind this phenomenon. However, as shown in Figure 6K-6M, the total F actin content was very low upon ERK3 knockdown, though some of these cells exhibited irregular protrusions and a reduction in polarized migration.

8. Do the authors see any effect with the knockdown of ERK4 as they are closely related?

We have indeed performed ERK4 knockdown experiments. However, we found that ERK4 might be regulating the stability of ERK3 in a cell type-dependent manner which needs to be investigated further. We also presented some of these observations in our previous work (Bogucka et al., *eLife* 2020 – please see Figure Suppl. 5 C-E). We would like to extend these observations and publish it as a separate study. Notably, they did not influence the conclusions of the current manuscript.

9. ERK3 is activated by PAK kinases (through phosphorylation at Ser 189) which are downstream of Rac1 and Cdc42 and the authors can add further discussion in these lines in the context of the summary figure presented in figure 8.

Good point. As proposed by this reviewer, we have added a short discussion about the possibility of a positive feedback loop, in which ERK3 is activated by EGF to stimulate the GTP loading of Cdc42 and Rac1 and their interaction with PAKs, which in turn phosphorylate and activate ERK3 (please see page 16 first paragraph).

10. Figure 7 is rather important and could be better utilised to bring clarity to their investigation. Moving the Rac1/Cdc42 activity data from Figure 7 -suppl2 would add important functional information. This should be complemented with a blot for pARP3-S418 in the expression of empty vector/ERK3 WT/ERK3 KD conditions. Together this would strengthen the authors' conclusions in the text.

We have moved Figure 7 Supplement 2 to the main figures now (Figure 7 E-I). In addi2on, we have probed the membranes from an experiment in which we have analyzed shERK3 cells complemented with wild type and kinase-dead versions of ERK3 and consistently, we found that this phospho site is clearly regulated in a kinase-dependent manner. These data are now included in the Supplement (please see Figure 6, Supplement 1 B).

11. In Figure 3F the authors show a convincing GEF assay with the ERK3 kinase domain and Cdc42. The experiment shown in Figure supp 1 has a very high starting fluorescence, can the authors explain what is going on here?

Thanks for this comment. We have performed the assay as per the manufacturer’s instructions (h=ps://www.cytoskeleton.com/pdf-storage/datasheets/bk100.pdf). Similar to the representative examples in the brochure, we detect different staring fluorescence which could be attributed to slight differences in starting temperature, GEF concentrations etc. However, despite these variations in settings, there is always a clear, dose-dependent increase in GEF activity upon addition of DBS or ERK3.

12. In Figure 4, the colocalization data is supportive, but there are no real controls. The authors should consider changing their wording on line 280 in the results text, from "corroborated" to "supported".

The wording has been changed as suggested.

13. It seems that none, or very little, of the overexpressed ERK3 (WT or mutant), is located in the vicinity of the actin cytoskeleton at the periphery of the cell. The authors should comment on this.

We agree that exogenous ERK3 doesn’t co-localize or localized in the vicinity of the actin cytoskeleton. We have commented on that observation in the manuscript: “We observed that exogenous ERK3 localized predominantly in the cytosol as compared to the endogenous ERK3, which we detected at the cell edges and F-actin rich protrusions (Figure 1A). These results prompted us to prefer ERK3 knockdown cells for the study of ERK3-mediated phenotypes. “.

The discrepancies in the localisation of overexpressed and endogenous proteins can be explained by the conformational changes which might have occurred due to overexpression of the exogenous construct. Additionally, one cannot rule out the possibility of changes in the stoichiometry and composition of ERK3 containing multimeric protein complexes. However in IP assays and in complementation experiments we could detect tagged ERK3 precipitating with the other interactors and rescue the phenotype in KD cells. Perhaps one should explore a knock in cell line with tagged ERK3 in the C-terminus and perform further studies. We hope that we can address these issues in the future.